# Structure of the p53 degradation complex from HPV16

John C. K. Wang [1,2], Hannah T. Baddock[1,2], Amirhossein Mafi[1], Ian T. Foe [1], Matthew Bratkowski [1], Ting-Yu Lin[1], Zena D. Jensvold [1], Magdalena Preciado López[1], David Stokoe[1], Dan Eaton [1], Qi Hao [1,3] ✉ & Aaron H. Nile [1,3] ✉

Human papillomavirus (HPV) is a significant contributor to the global cancer burden, and its carcinogenic activity is facilitated in part by the HPV early protein 6 (E6), which interacts with the E3-ligase E6AP, also known as UBE3A, to promote degradation of the tumor suppressor, p53. In this study, we present a single-particle cryoEM structure of the full-length E6AP protein in complex with HPV16 E6 (16E6) and p53, determined at a resolution of ~3.3 Å. Our structure reveals extensive protein-protein interactions between 16E6 and E6AP, explaining their picomolar binding affinity. These findings shed light on the molecular basis of the ternary complex, which has been pursued as a potential therapeutic target for HPV-driven cervical, anal, and oropharyngeal cancers over the last two decades. Understanding the structural and mechanistic underpinnings of this complex is crucial for developing effective therapies to combat HPV-induced cancers. Our findings may help to explain why previous attempts to disrupt this complex have failed to generate therapeutic modalities and suggest that current strategies should be reevaluated.

Human papillomaviruses (HPVs) are composed of more than 200 known HPV subtypes which can infect human skin and mucous membranes[1]. High-risk HPV subtypes, including HPV16 and HPV18, are associated with the development of various cancers such as anal, vaginal, vulvar, penile, oropharyngeal and, infamously, ~90% of cervical cancers[2]. The link between HPV and human cancer was first implicated in the 1980s when HPV DNA was identified in cervical cancer tissue[3,4]. Subsequent epidemiological and molecular studies demonstrated a strong association between the expression of HPV oncoproteins and the development of HPV-associated cancers[2,5,6]. Clinical trials later revealed that prophylactic HPV-vaccines are highly effective at preventing development of these cancers, further supporting the causal relationship between HPV and cancer[7].

The discovery of the HPV early protein 6 (E6) and early protein 7 (E7) oncoproteins was a breakthrough in understanding HPV-driven cancers. The development of carcinogenesis through high-risk HPV subtypes involves the expression of two viral oncoproteins: E6 and E7.

These interfere with the normal functions of cellular tumor suppressor proteins, p53 and Rb, respectively[8]. Multiple lines of evidence support the role of HPV16 E6 (16E6) and HPV18 E6 (18E6) as key molecular drivers of carcinogenesis. In vitro studies demonstrate that expression of 16E6 or 18E6 transforms human cells[9–11] and promotes tumorigenicity in mouse tissues including the cervix, skin, and oral mucosa[12]. Additionally, HPV E6 oncoproteins not only have direct effects on cellular proliferation and apoptosis but also alter numerous other cellular processes, such as DNA damage repair, cell signaling, and gene expression, which may also contribute to their oncogenic properties[13].

HPV E6 proteins are approximately 150 amino acids in length (Supplementary Fig. 1) and interact with multiple host proteins[14], including the E3-ligase, E6AP, which is also known as UBE3A[15,16]. Recruitment of E6AP by 16E6 forms a neo-destruction complex that promotes p53 degradation through E6AP-mediated polyubiquitination, leading to uncontrolled cell proliferation and tumorigenesis[17–20]. Although nuclear magnetic resonance (NMR) of 16E6 solved the

[1]Calico Life Sciences LLC, 1170 Veterans Blvd, South San Francisco, CA 94080, USA. [2]These authors contributed equally: John C. K. Wang, Hannah T. Baddock. [3]These authors jointly supervised this work: Qi Hao, Aaron H. Nile. ✉e-mail: qhao@calicolabs.com; aaronnile@calicolabs.com

isolated 16E6 N- and C-terminal lobes, it failed to resolve the full-length structure due to poor protein behavior[21–23] (Supplementary Fig. 2a). Subsequent mutagenesis of four cysteine residues to serine improved 16E6 solubility and homogeneity (referred to as 4C4S), enabling the crystallization of HPV16 E6 in complex with the E6AP LXXLL peptide[24] (Supplementary Figs. 1, 2b). Later, the p53 core domain (p53$^{core}$) was incorporated into the complex, providing insight into the structural rationale for 16E6-mediated degradation of p53 [25] (Supplementary Fig. 2c).

Taken together, previous studies revealed a dumbbell-shaped 16E6 protein composed of N- and C-terminal lobes each with a Zn-finger and connected through an α-helical linker peptide. In silico simulations of the apo-16E6 protein[26–28] and NMR[23] predict that the 16E6 N- and C-terminal lobes are flexible relative to each other, providing a rationale for the challenges in obtaining apo-16E6 crystals. A recent apo-16E6 structure was deposited to the PDB which has an elongated and distorted alpha-helical linker between the N- and C-terminal lobes; however, the publication describing this structure is yet to be released (PDB ID no 7UAJ). The E6AP LXXLL peptide occupies and stabilizes the N- and C-terminal 16E6 lobes through binding at a hydrophobic groove via interactions dominated by the E6AP leucine triad in L(409)XXLL(413)[24,29] (Supplementary Fig. 2d). Little is known about interactions between E6AP and 16E6 beyond the LXXLL peptide which accounts for only 1.4% of the E6AP primary amino acid sequence (Supplementary Fig. 2e).

In this work, we build upon previous research, including recent yeast two-hybrid studies that suggested alternative 16E6 and E6AP interactions beyond that known for the E6AP LXXLL binding domain[30], but lacked structural and biophysical validation. Prior studies have employed various methods such as molecular dynamics simulations[31], mass spectrometry[32], and chemical biology approaches[33], yet they often relied on large protein truncations. In this work, we advance the understanding of this complex by solving the cryoEM structure of the full-length 16E6, E6AP, and the p53$^{core}$ domain at ~3.3 Å resolution. Our findings unveil an extensive interaction interface between 16E6 and E6AP, characterized by high-affinity picomolar binding between the two proteins. This discovery provides a structural basis for the observed picomolar interaction and offers insights into the dynamics of this complex. Our mutagenesis and ubiquitination assays coupled with molecular dynamics simulations further corroborate these interactions, highlighting their functional significance in HPV-driven carcinogenesis. By exploring these aspects of the HPV16 E6-mediated p53 degradation, our results not only enhance the understanding of this mechanism, but also provide a plausible rationale for the failure of past drug discovery efforts targeting the 16E6 and E6AP protein-protein interaction interface.

## Results

### HPV16 E6 binds to E6AP with picomolar affinity

The binding affinity between full-length E6AP and an HPV E6 protein has not been previously reported. To evaluate this interaction, we obtained pure and homogeneous full-length human E6AP expressed from insect cells and MBP-16E6(4C4S) expressed from E. coli (Materials and Methods and Supplementary Fig. 3) and experimentally tested the binding affinity using multiple techniques. In vitro measurement of the binding affinity by surface plasmon resonance (SPR) between MBP-16E6(4C4S) and biotinylated E6AP using CAP sensor chips revealed a $K_D$ of 144 pM (Fig. 1a), similar to the $K_D$ of 206 pM using streptavidin (SA) sensor chips (Supplementary Fig. 4). To further validate this interaction, label-free mass photometry determined the MBP-16E6 (4C4S) and E6AP to form a 1:1 stoichiometry complex (Supplementary Fig. 5a, b) also with measured $K_D$ values ranging between 97 pM to 1.29 nM (Supplementary Fig. 5c–e). In addition, we developed a recombinant NanoBRET system using MBP-16E6-HaloTag® protein as an acceptor and E6AP-Nanoluc as a donor (Supplementary Fig. 6) to measure in solution binding through proximity energy transfer. Using NanoBRET we found a ~96-fold reduction in the $K_D$ (measured at 15.27 nM). We reasoned that the reduction is due to the interference from the large Nanoluc and Halo tags as interaction between MBP-16E6-Halo and E6AP-Nanoluc is measured at 4 nM by SPR (~20-fold lower affinity compared with matched proteins without NanoBRET tags, Supplementary Fig. 7). Normalizing for the reduced affinity, the variation between NanoBRET and SPR was approximately 5-fold. This data in combination with the observed subnanomolar $K_D$ by mass photometry support the ability of SPR to faithfully capture the high affinity interaction between MBP-16E6 and E6AP.

Compared to previously reported binding constants, we measured an approximately 10,000-fold higher affinity than the 2−4 μM $K_D$ measured by SPR[34,35] for the isolated E6AP LXXLL peptide. Taken together, the high affinity interaction between 16E6 and full-length E6AP suggests extended interactions beyond those reported for the E6AP LXXLL peptide and 16E6. To investigate the structural basis of this interaction, we determined the structure of the 16E6, E6AP, and the p53$^{core}$ ternary complex using single-particle cryoEM.

### Structural determination of the 16E6, E6AP, and p53 degradation complex

We purified pure and homogeneous MBP-16E6(4C4S) and the p53$^{core}$ domain from E.coli (Supplementary Fig. 3). To aid in structural studies, we adjusted the linker between MBP and 16E6 to be more rigid than the one used in SPR experiments (Supplementary Fig. 3a). After incubating MBP-16E6, E6AP, and the p53$^{core}$ domain, we isolated the ternary complex using size exclusion chromatography (Fig. 1b). We then used cryoEM to image the complex and determined the overall resolution to be ~3.3 Å (Fig. 1c and Supplementary Fig. 8). The protein-protein interaction interfaces have a higher resolution compared to the solvent-exposed regions, likely due to the tight inter-chain interactions. Using the reported crystal structures of 16E6 and the p53$^{core}$ domain coupled with the E6AP AlphaFold2 model, we refined the domain placement within the electron density map which revealed that 16E6 is sandwiched between E6AP and p53$^{core}$ proteins (Fig. 1d).

We modeled the 16E6, E6AP, and p53$^{core}$ structures into the electron density, revealing a 'Y'-like conformation, with E6AP and p53$^{core}$ flanking HPV16 E6 (Fig. 2a, b) with the individual 16E6 and p53 proteins superimposing well onto previous structures (Supplementary Fig. 9). The interaction interface between 16E6 and E6AP is significantly larger in the ternary complex compared to the 939 Å$^2$ LXXLL peptide interface which expanded to approximately 2,361 Å$^2$, or ~2.5-times larger, contributing approximately −13.23 kcal/mol binding energy (Supplementary Fig. 10a,b). In contrast, the interaction interface between 16E6 and p53$^{core}$ is similar to previous reports[25] at 1,705 Å$^2$ contributing approximately −6.1 kcal/mol to the binding energy (Supplementary Fig. 10c). Overall, the 16E6 structure resembles a hand that connects E6AP and p53 through a saddle-shaped interaction network of 3,972 Å$^2$, with an approximate −17.2 kcal/mol interaction energy. This unreported large interface explains the structural basis of the observed high-affinity, picomolar binding affinity between 16E6 and E6AP (Fig. 1a, Fig. 2a, b and Supplementary Fig. 10d).

Approximately 65% of the E6AP primary amino acid sequence of E6AP spanning from T126−G761 was resolved (Supplementary Fig. 11). However, regions that include the N-terminus (M1−V125), K171−V232, V388−E394, and G433−D438 were not resolved, largely due to predicted flexibility within these regions (Supplementary Fig. 11 and Supplementary Fig. 12a, b). Utilizing comparison of our structure to the AlphaFold2 E6AP model, we classified these regions as elongated and flexible loops (Supplementary Fig. 13a–c). The C-terminal lobe of the HECT domain (residues G761−Y852), which is essential for E6AP's enzymatic function, was not resolved in our structure (Supplementary Figs. 11, 13). Nevertheless, the C-terminal HECT domain structure of human E6AP has been previously reported[36]. Comparing the

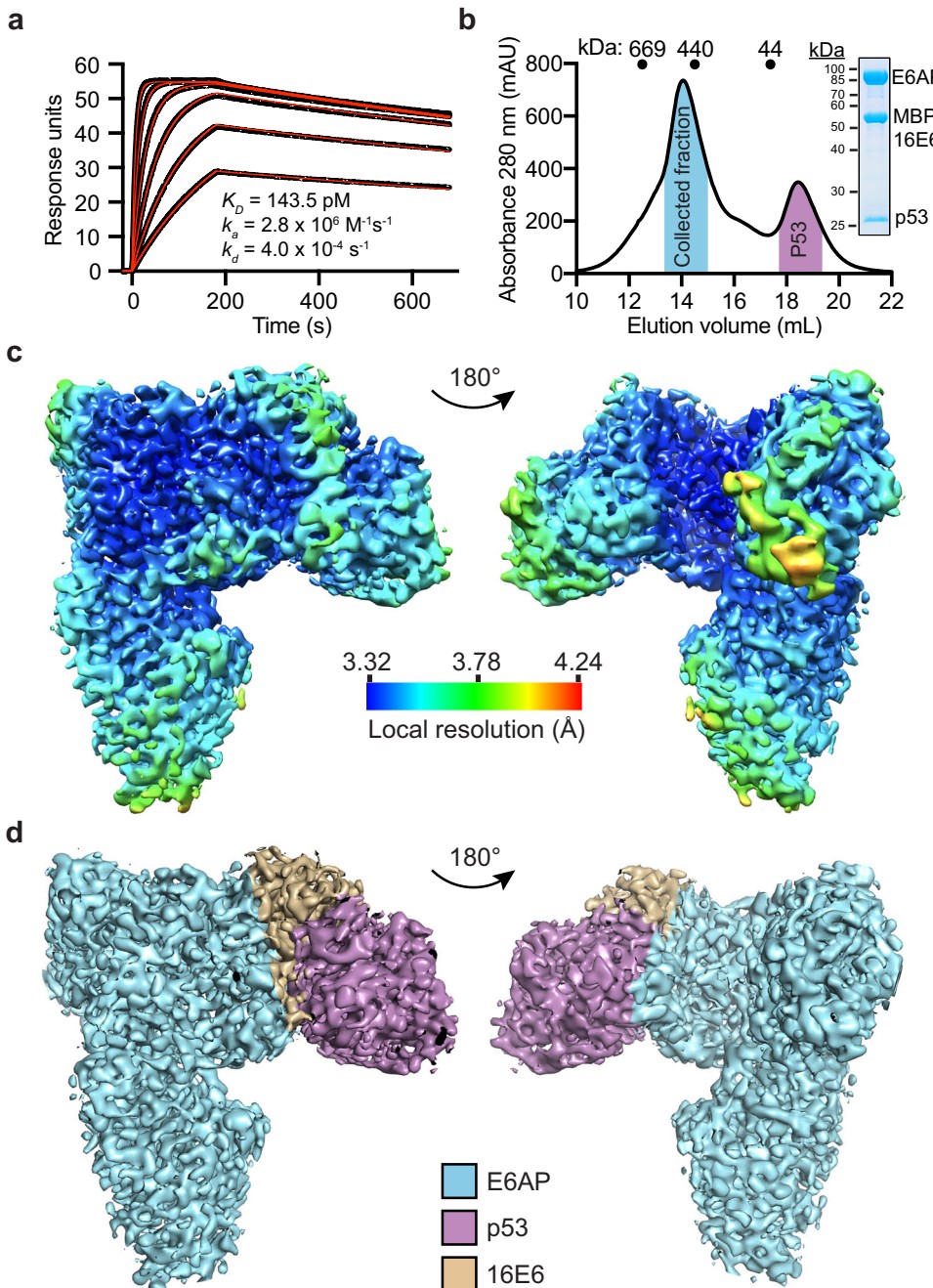

**Fig. 1 | CryoEM density map of the ternary complex formed by HPV16 E6 (16E6), full length E6AP and the p53^core domain. a** Surface plasmon resonance (SPR) measurements of MPB-16E6(4C4S) binding to biotin-E6AP at varying concentrations are shown. The black line represents the mean binding response of three independent experiments, while the red line represents the curve fit to a 1:1 interaction model. $K_D$, dissociation constant; $k_a$, association rate, $k_d$, dissociation rate. **b** The MBP-16E6, E6AP, and p53^core ternary complex was resolved and collected by size exclusion chromatography using a Superdex 200 column. The trace is representative of two independent experiments. The collected fraction which contains the ternary complex is highlighted in blue, while the p53 fraction is in purple. An insert shows the ternary complex resolved by SDS-PAGE and stained with Coomassie blue. **c** The electron density map of the MBP-16E6, E6AP, and p53^core ternary complex is shown, colored according to the local resolution across a color gradient. High resolution areas are shown in blue, while low resolution areas are shown in red/yellow. **d** The density map from (**c**) is colored according to the location of each ternary complex member. HPV16 E6 is shown in tan, E6AP is shown in blue, and p53^core is in purple.

C-terminal region with the AlphaFold2 model, the structured HECT domain is connected to the rest of the protein through a potentially flexible unstructured region, which could account for the lack of resolution of this domain (Supplementary Fig. 12). To determine whether the flexibility of the C-terminal lobe of the HECT domain was the primary factor preventing its resolution by cryoEM, we carried out a ~400 ns MD simulation on the apo-E6AP structure predicted by AF2 (Supplementary Fig. 12b–f). Our analysis revealed that the C-terminal lobe of the HECT domain exhibits substantial deviation from the initial AF2 conformation, with an RMSD value of 11.5 ± 2.9 Å, measured for the Cα atoms, leading to significant flexibility (up to 10 Å) within this region (Supplementary Fig. 12b, d).

**Analysis of the HPV16 E6 and E6AP interaction interface**
Although 16E6 and E6AP form a contiguous interface, we divided the interaction interface into three segments, labeled site 1, site 2, and site

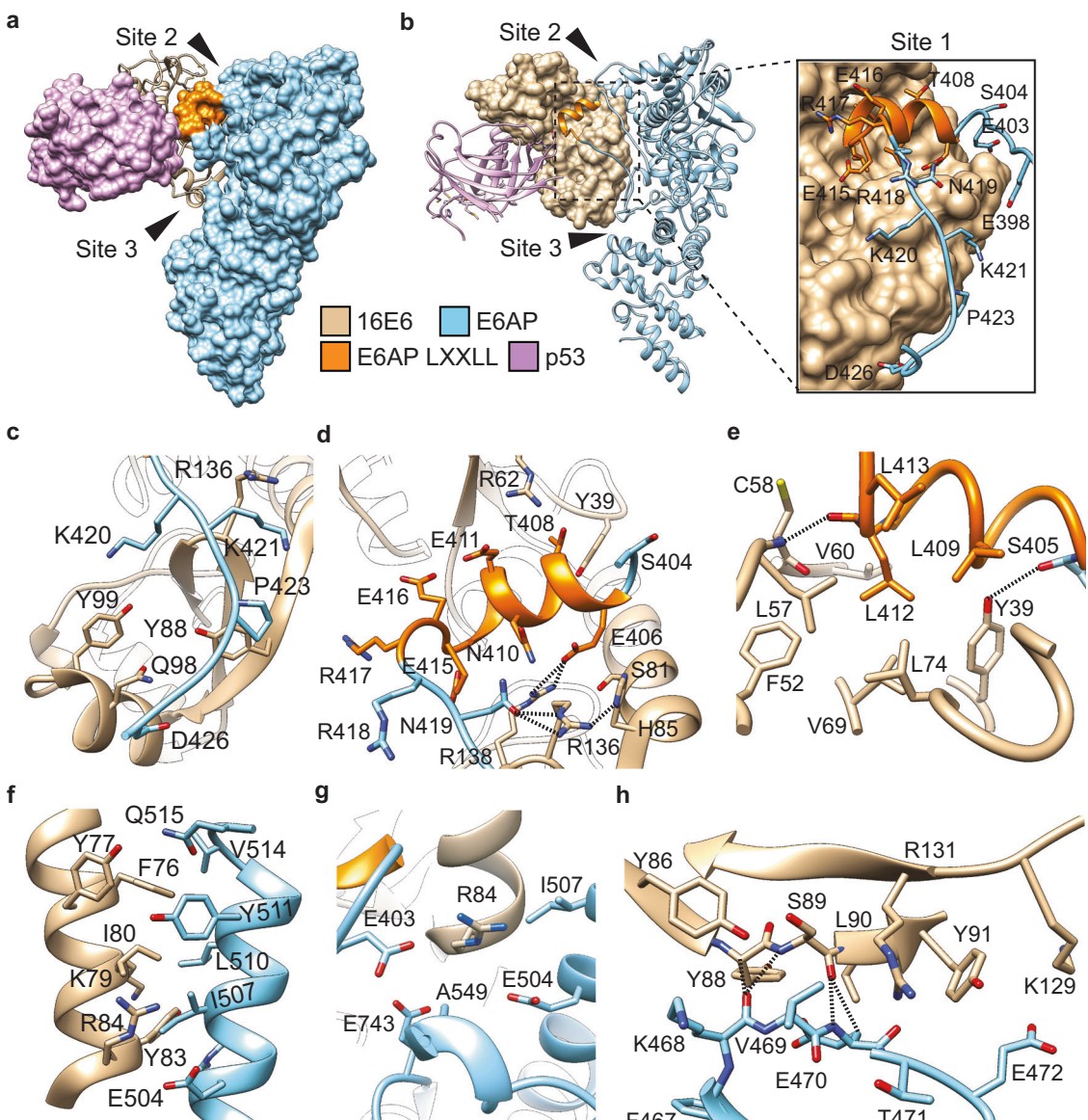

**Fig. 2 | Three primary sites drive 16E6 and E6AP interaction. a** Surface representation of E6AP (blue) in complex with 16E6 (tan, ribbon representation) and the p53[core] domain (purple, surface representation), highlighting the reported E6AP LXXLL motif (orange, surface representation). **b** Model from (**a**) with E6AP (blue, stick/ribbon representation), 16E6 (tan, surface representation), and p53[core] (purple, ribbon representation), with the E6AP LXXLL motif shown in orange (ribbon representation). A zoomed-in view of site 1 residues highlighting the side chains of E6AP (blue/orange, stick/ribbon representation) is also shown. **c**–**e** Site 1 residues that form the interaction interface between E6AP (blue, stick/ribbon representation) and 16E6 (tan, stick/ribbon representation), highlighting hydrogen bonding and ionic interactions (black, dashed lines) are depicted. **f** Site 2 interactions between 16E6 (tan, sick/ribbon representation) and E6AP (blue, stick/ribbon representation) primarily composed of van der Waals interactions. **g** Interaction between 16E6 R84 at the site 1 and site 2 interface. **h** Site 3 interactions highlighting backbone hydrogen bonding interactions (black, dashed lines) between 16E6 (tan) and E6AP (blue).

3 (Fig. 2 and Supplementary Fig. 11). Site 1 (between E6AP residues E403–D426) forms an interaction interface of ~1343 Å² and contributes approximately −7.78 kcal/mol of binding energy. The protein-protein interactions at site 2 (E6AP residues S503–Q515) produce an interface of approximately 670 Å² and contribute approximately −5 kcal/mol binding energy. Site 3 interactions (E6AP F467–N473), form an interaction-interface of approximately 500 Å² and contribute approximately −4.3 kcal/mol of binding energy. In total, the combination of site 1 through site 3 interactions generate a large interface of approximately 2,361 Å² between 16E6 and E6AP, contributing approximately −13.2 kcal/mol binding energy (Supplementary Fig. 10b).

Site 1 is composed of E6AP residues, E403 to D426, including the reported E6AP LXXLL motif (residues E406–E415); however, in our

structure, the LXXLL motif is protracted to encompass E395 and E431 which extends from, and is perpendicular to, the 16E6 hydrophobic groove (Fig. 2b). Site 1 interactions contain multiple contacts including van der Waals, hydrogen, and ionic bonds formed by E6AP residues from E403 to D426 (see Fig. 2b–e). Notably, in the region between G414 and K420, distal to the LXXLL motif, numerous van der Waals interactions between E6AP residues K420, K421, G422, P423, D426 and 16E6 residues Y88, Q98, Y99, R136 and G137 were observed (Fig. 2c). Between S404–N419 a loop turns into the LXXLL alpha-helix and reverts into a loop (Fig. 2d). Within this region, the carboxylic acid of E6AP N419 forms a hydrogen bond with the primary amine and guanidino nitrogen of 16E6 R136, which is partially stabilized by a hydrogen bond to the imidazole nitrogen of 16E6 H85 (Fig. 2d). Additionally, E6AP E406 forms hydrogen bonds with 16E6 R138 and van der Waals

interactions with several residues displayed in Fig. 2d. The core LXXLL α-helix between E6AP residues L413 and S405 is constrained by hydrogen bonds between the backbone amine of 16E6 C58 and the backbone carbonyl of E6AP L412, and between the backbone carbonyl of E6AP S405 and the hydroxyl group of 16E6 Y39 (Fig. 2e). Leucine residues within the LXXLL motif, L409, L412, and L413 form van der Waals interactions across the floor of the 16E6 hydrophobic groove formed in part by Y39, F52, L57, V60, V69, L74, Y77, S78, S81, R109 and R138 (Fig. 2d, e).

The site 2 interface is formed between the α-helical linker between the N- and C-terminal 16E6 lobes and the E6AP α-helix between residues S503–Q515. This interface consists of van der Waals and ionic interactions. Specifically, 16E6 F76, Y77 and I80 form van der Waals contacts with E6AP I507, L510, Y511, V514, and Q515, which together form a shallow interlocking structure surrounding 16E6 F76 (Fig. 2f) whereas 16E6 Y83 and K79 interact with S503, I507 and L510 through additional van der Waals interactions. Ionic interactions between 16E6 R84 and E6AP E504 were also observed (Fig. 2f). Interestingly, 16E6 R84 is located between site 1 and site 2, and forms ionic interactions with E6AP E403 and E504 in addition to van der Waals interactions with E6AP E403, E504, I507, A549, and E743 (Fig. 2g). The site 3 interface occurs between the C-terminal lobe of 16E6 and a β-strand directly after the α-helical linker connecting the N- and C-terminal lobes which form multiple van der Waals and hydrogen bonding interactions. Specifically, 16E6 Y88 and 16E6 S89 form hydrogen bonds with E6AP K468 and 16E6 S89 also forms hydrogen bonds with E6AP E470 (Fig. 2h). Finally, multiple van der Waals interactions are observed between 16E6 Y86, Y88, S89, L90, Y91, K129, R131 and E6AP residues P423, D426, F467, K468, V469, E470, T471, and E472 (Fig. 2h).

To further understand how the three interaction sites may impact the E6AP binding to 16E6, we employed two replicate long-time scale MD simulations on the 16E6, E6AP and p53[core] ternary complex (see Materials and Methods). Our analysis of the RMSD for each protein in the complex indicates that the entire construct remains relatively stable without significant structural changes during the simulation (Fig. 3a, right). Notably, the E6AP LXXLL motif, specifically residues E406–E415 in site 1, remains highly stable and maintains its alpha-helical structure with an RMSD value of 0.4 ± 0.2 Å, suggesting an important role of this LXXLL-motif mediated interface (Fig. 3a, b, right; Supplementary Fig. 14a). On the other hand, flexibility in the LXXLL extended region was observed, including E393–E431 (RMSD = 4.5 ± 0.8 Å in Replicate 1) in the site 1 interface, suggesting the E6AP-16E6 interaction is also dynamic in nature (Fig. 3a, right Supplementary Fig. 14a). Nevertheless, our MD-simulation reveals that the extended LXXLL motif remained close to 16E6 throughout the simulation and the distance was measured within 11 Å (Fig. 3b, left; Supplementary Fig. 14a), demonstrating the stability of 16E6-E6AP despite the dynamics in certain regions. In addition, the interface between E6AP and 16E6 at site 2 also remains stable around the distance of 12.0 Å (Fig. 3c, left; Supplementary Fig. 14b). However, the interface at site 3 exhibits a higher degree of dynamism, fluctuating between 15.0 Å and 28.0 Å of the 16E6, before eventually returning to 20.0 Å (Fig. 3d and Supplementary Fig. 14c), indicating that different from sites 1/2, site 3 is unlikely to be the primary site of interaction between E6AP and 16E6.

The superimposition of the E6AP AlphaFold2 model with our E6AP structure revealed a conserved overall fold (Supplementary Fig. 13b). However, comparison of the AF2 E6AP structure and our cryoEM structure reveal low RMSD values except a 16.5 Å shift at L415 within the E6AP LXXLL motif (Supplementary Fig. 13b, c). The Alpha-Fold2 prediction showed a distorted LXXLL alpha-helical structure in the apo-E6AP structure, which could be attributed to its low predicted pLDDT score (Supplementary Figs. 12b, c, 13). This unstructured configuration allows us to consider that the LXXLL motif is highly flexible in the isolated E6AP state. We hypothesize that the well-defined alpha-helical structure of the motif is stabilized by its complex formation

with 16E6. Thus, we examined this hypothesis by performing additional MD simulations (~700 ns on aggregate).

Starting with the AF2 apo-E6AP protein, root mean square fluctuation and clustering (Fig. 4a, Supplementary Figs. 12a, b, and 15a) of the entire trajectory of E6AP conformation during the 400 ns MD simulation shows that the E6AP LXXLL motif features a highly flexible character, supporting our hypothesis that the LXXLL motif is highly flexible in the absence of 16E6. However, to eliminate the possibility that the high degree of flexibility of the LXXLL motif is solely due to the apo-E6AP construct and not to poor prediction by AlphaFold 2, we performed a separate 200 ns MD simulation of the apo-E6AP protein using the cryoEM E6AP, 16E6, and p53[core] complex as a starting point. We removed the entire 16E6 and p53[core] protein to create an apo-E6AP protein, which was then immersed in ~56 K water molecules and relaxed at 310 K. Results show that the E6AP LXXLL motif, which adopted an alpha-helix structure in the ternary complex, became highly flexible, consistent with our findings for the apo-E6AP protein from AlphaFold 2 (Fig. 4b). Collectively, these results support the high flexibility of the isolated E6AP LXXLL motif due to its lack of binding to 16E6.

To further evaluate the impact of 16E6 on the stability of the E6AP LXXLL motif, we conducted two additional MD simulations. In one simulation, we started with the ternary cryoEM complex and removed the entire p53[core] protein (Fig. 4c). The resulting 16E6-E6AP complex was immersed in 61 K water molecules and relaxed for ~250 ns at a temperature of 310 K to assess the possible impact of p53[core] on the stability of the E6AP motif. In the second simulation, we removed the entire E6AP protein except for the LXXLL motif region, I401–N419 (Fig. 4d). The resulting complex was solvated with 23 K water molecules and relaxed at 310 K for 200 ns to address or eliminate the possible impacts of the whole E6AP protein on the stability of the LXXLL motif. Both MD simulations demonstrated that the presence of 16E6 stabilized the isolated LXXLL motif, reducing its fluctuation.

To further probe the stability of the 16E6 and E6AP complex, we employed an engineered E6AP LXXLL-derived covalent peptide (and thus competitive binder with E6AP), peptide-13 (pep13), with an apparent Ki of 17 nM[37]. Incubation of pep13 over 3-days did not modify MBP-16E6 in the presence of E6AP; however, pep13 but not the inactive derivative 3L3A peptide formed a covalent adduct with MBP-16E6 in isolation (Supplementary Fig. 16) supporting the observed high affinity interaction between 16E6 and E6AP.

## Rationally designed mutations disrupt the 16E6 and E6AP interaction

To confirm the functional role of the observed 16E6 and E6AP interaction interface, we rationally designed mutations in 16E6 that are predicted to block complex formation and thus inhibit in vitro p53 ubiquitination. All mutants were generated in the 16E6 4C4S background, purified to homogeneity and their protein integrity was confirmed using LC-MS, SDS-PAGE, and analytical size exclusion chromatography (Supplementary Fig. 17 and Supplementary Fig. 18).

First, we assessed the reported 16E6 L57A mutant, which disrupts key hydrophobic interactions at site 1 and blocks MBP-LXXLL peptide binding to 16E6[24,25]. We immobilized biotinylated E6AP onto SPR sensor chips and flowed over MBP-16E6 or mutant protein to assess the affinity of each to E6AP. In this assay, we observed minimal binding of 16E6 L57A to E6AP, whereas MBP-16E6 has a binding affinity of $K_D$ = 144 pM (Fig. 1a and Fig. 5a). At site 1, MBP-16E6 R62 forms van der Waals interactions with E6AP T408 and E411 and its mutation to R62A modestly reduces the binding affinity to E6AP by 16-fold ($K_D$ = 2.2 nM). Mutation of MBP-16E6 at R84 to either Ala or Glu is predicted to interfere with both site 1 and site 2 recognition and strikingly reduced the binding affinity by ~239-fold ($K_D$ = 34.4 nM) and ~1355-fold ($K_D$ = 195 nM), respectively (Figs. 2g and 5a). The site 2 mutation 16E6 F76A ($K_D$ = 8.5 nM) reduced the binding affinity to E6AP by 59-fold

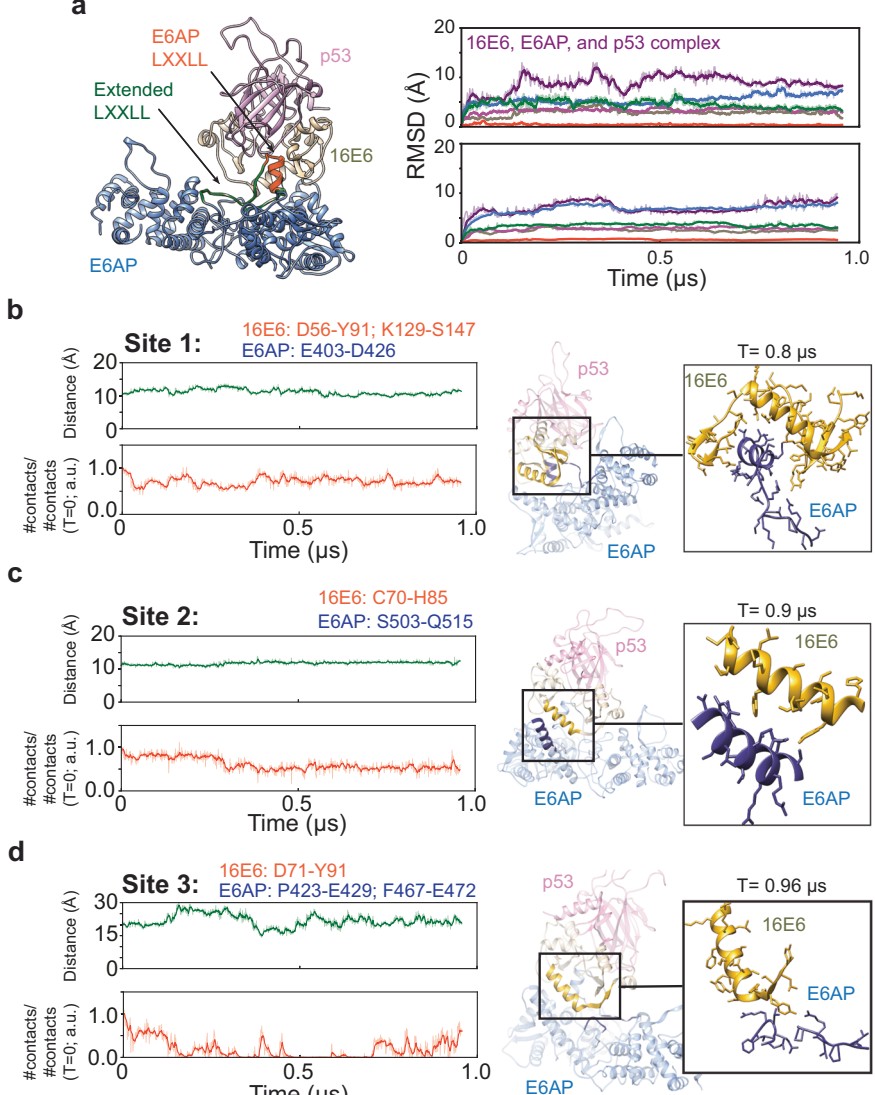

**Fig. 3 | Molecular dynamics simulations of the p53, 16E6, and E6AP ternary complex. a** Left: the optimized 16E6-E6AP-p53 ternary complex after ~1 μs simulation. Right: Root mean square deviation (RMSD) trajectory analysis for each protein in the ternary complex, showing overall complex stability without significant structural changes. **b** The E6AP LXXLL motif (E406-E415) in site 1 retains its alpha-helical structure with an RMSD value of 0.4 ± 0.2 Å (replicate 1), underscoring its interface role. Stability analysis of the E6AP-16E6 interactions at site 1, measuring the center of mass distance between the Cα atoms of E6AP (residues E403-E426) and 16E6 (residues D56-Y91, K129-S147). Despite some LXXLL extended region flexibility, the motif remains proximal to 16E6, reinforcing 16E6-E6AP stability.

**c** Stability analysis of the E6AP-16E6 interactions at site 2, examining the center of mass distance between the Cα atoms of E6AP (residues S503-Q515) and 16E6 (residues C70-H85). The interface remains stable at approximately 12.0 Å. **d** Stability analysis of the E6AP-16E6 interactions at site 3, gauging the center of mass distance between the Cα atoms of E6AP (residues P423-E429, F467-E472) and 16E6 (residues D71-Y91). Site 3 exhibits higher dynamism, fluctuating between 15.0 Å and 28.0 Å, highlighting it as a non-primary interaction site. Contact fraction computation utilized the 'gmx mindist' module within the GROMACS software suite. Replicate experiments are found in Supplementary Fig. 14. "a.u" stands for arbitrary unit.

likely through disruption of van der Waals interactions (Figs. 2f and 5a). However, mutation of MBP-16E6 Y77A ($K_D = 742$ pM) modestly reduced binding to E6AP by ~5-fold, likely through the disruption of van der Waals interactions between Y511 and Q515 on E6AP. The site 3 mutants, Y86A, Y88A, and S89A, caused a reduction in the association between 16E6 and E6AP, resulting in an 11- to 38-fold lower affinity with measured $K_D$ values of 6.3 nM, 5.4 nM, and 1.71 nM, respectively (Figs. 2h and 5a). However, accurately interpreting the underlying cause of reduced binding affinity for site 3 mutations is confounded by the limited side chain interactions, and their effects are likely due to indirect effects (Fig. 2h). As an orthogonal approach we confirmed that a similar trend in reduced activity holds for the above mutations when incorporated into the solution based NanoBRET system using mutant or WT MBP-16E6-Halo and E6AP-Nluc (Supplementary Fig. 19).

Nonetheless, these mutations validate the observed interactions and provide tools to evaluate the functional consequences of disrupting the 16E6 and E6AP complex biochemically.

To evaluate the effects of mutations disrupting the interaction between 16E6 and E6AP on their ability to polyubiquitinate p53, we performed in vitro p53 ubiquitination assays. In this assay, we incubated E1 (100 nM), E2 (200 nM), E6AP (200 nM), MBP-16E6 (200 nM), full-length p53 (200 nM) and initiated the ubiquitination cascade by introducing Mg-ATP into the reaction which terminates at p53 or E6AP. After the reaction was complete it was resolved by SDS-PAGE and probed for p53 or E6AP by western blot. The assay was validated using MBP-16E6(4C4S), demonstrating robust polyubiquitination of p53 and E6AP (Fig. 5b). Previous reports have also demonstrated that the presence of 16E6 promotes the auto-ubiquitination of E6AP[38], consistent

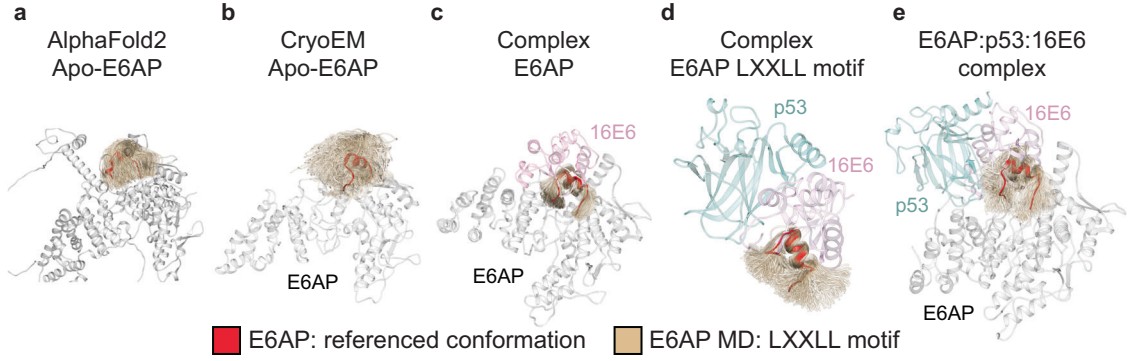

**a** AlphaFold2 Apo-E6AP   **b** CryoEM Apo-E6AP   **c** Complex E6AP   **d** Complex E6AP LXXLL motif   **e** E6AP:p53:16E6 complex

■ E6AP: referenced conformation   ■ E6AP MD: LXXLL motif

**Fig. 4 | Molecular dynamics simulations of apo or complexed E6AP-LXXLL motifs.** Different states of the E6AP LXXLL motif peptide (tan), determined by clustering the entire MD trajectories (refer to Materials and Methods). The parental LXXLL motif conformation is highlighted in red, featuring both the AlphaFold 2 (left) and cryoEM conformations (all other panels). Initial configurations for the MD simulation are as follows: **a** apo-E6AP protein as predicted by AF2, **b** apo-E6AP after removing p53 and 16E6 from our CryoEM complex, **c** binary complex of 16E6-bound E6AP, derived from the cryoEM structure with p53 removed, **d** ternary complex of p53-16E6-E6AP-LXXLL motif, using only the mentioned section and omitting the rest of the E6AP construct from the cryoEM and **e** ternary complex of p53-16E6-E6AP as observed in our cryoEM.

with our observations. We then confirmed that the signal for both p53 and E6AP were within the assay's linear range allowing for accurate quantification of the reaction (Supplementary Fig. 20a, b). After normalizing to the no-ATP condition to account for differences in protein loading, we quantified the extent of p53 ubiquitination or E6AP autoubiquitination for each condition as determined by the proportion of unmodified p53 or E6AP remaining. MBP-16E6 promoted p53 ubiquitination, resulting in ~13% of unmodified p53 remaining after 15 minutes, and the reaction then plateaued with ~4.5% of unmodified p53 remaining at 30 min. Similarly, MBP-16E6 promoted E6AP autoubiquitination, with ~28% of unmodified E6AP remaining at 15 min., ~8.7% at 30 min., and the reaction then plateaued at ~5.4% by 45 min. (Fig. 5b-e and Supplementary Figs. 20, 21). The reaction kinetics differed slightly between p53 and E6AP ubiquitination; however, the data show that ~90–95% of full-length p53 and E6AP are modified within 30 minutes, providing a robust signal window to evaluate the effects of mutant MBP-16E6 on E6AP-mediated p53 ubiquitination and E6AP autoubiquitination. Taken together, we chose to compare wild-type and mutant MBP-16E6 proteins at 30 min.

The MBP-16E6 L57A resulted in unmodified p53 (64.4%) and E6AP (45.6%) remaining after 30 min. (Fig. 5c–e and Supplementary Figs. 20, 21). The site 1 mutant R62A resulted in parental p53 (18.4%) and E6AP (9.3%) remaining, whilst for the site 1 and site 2 mutant, R84A, the proportion of unmodified p53 and E6AP, was 43.9% and 64.9%, respectively. For the charge reversal mutant, R84E, this effect was more pronounced for both p53 (77.2%) and E6AP (64.9%) (Fig. 5c–e and Supplementary Figs. 20, 21). At site 2, the F76A mutant exhibited 59.5% of unmodified p53 levels and 82% for E6AP at 30 min., whilst the Y77A mutant exhibited 22.4% of unmodified p53 and 21.6% E6AP. At site 3, the Y86A, Y88A, and S89A mutants had unmodified p53 levels of 44.2%, 19.2%, and 57.3%, respectively. For Y86A, Y88A, and S89A, the remaining unmodified E6AP levels at 30 min. were 56.5%, 21.2%, and 33.1% (Fig. 5c–e and Supplementary Figs. S20, S21). Taken together, there is good agreement between mutants that disrupt 16E6 and E6AP complex formation and a concomitant decrease in 16E6-mediated p53 ubiquitination and E6AP autoubiquitination. The results suggest that multiple interaction points are important for robust complex formation and subsequent p53 and E6AP ubiquitination.

**Recognition mechanism of p53 by the 16E6 and E6AP complex**
The cryoEM structure shows a similar interaction interface between p53 and 16E6 as the previously reported X-ray structure[25]. Specifically, the cryoEM p53[core] and 16E6 interaction interface consists of ionic and hydrogen bonding interactions and numerous van der Waals

interactions, forming an approximately 1705 Å² interface (Supplementary Figs. 10c and 22). However, due to the lack of resolution across several loops on p53[core] in the cryoEM structure (i.e. loops P223–D228, E198–R202, D259–N263 and E180–L188), a detailed analysis of this interface was not included. Zainer et al. reported interactions between p53[core] and the C-terminus of the MBP-LXXLL fusion protein used in their structural studies. However, the structural information was limited as the C-terminus terminates at R417 in chain A and E416 in chain B. This places the unconstrained C-terminus close to the p53[core] domain, forming a non-native hydrogen bond pair with p53 R110 (Supplementary Fig. 23). After resolving the cryoEM E6AP LXXLL region, hydrogen bonding and ionic interactions were observed between E6AP R417 and p53 D148 and R110 (Fig. 6a (i)).

To further clarify the molecular interactions that occur at the interface between E6AP and p53, we carried out two independent MD simulations. Our findings complement our cryoEM structure by revealing that in addition to R417, R418 also plays a crucial role in making direct contacts with p53. Specifically, R418 dynamically interacts with the D228 residue of p53 and interchangeably forms a salt bridge with D148, a role it shares with R417. This dynamic interplay allows R417 to frequently establish an ionic interaction with D228. Thus, our MD simulations predict that both R417 and R418 play significant roles in mediating the interactions between E6AP and p53 (Fig. 6a).

To assess the contribution of this interaction, we generated pure and homogeneous single and double E6AP mutants to selectively disrupt E6AP and p53 but not 16E6:E6AP interactions (Supplementary Fig. 24) and demonstrated that all mutants maintain robust interactions with 16E6 by SPR (Supplementary Fig. 25). To monitor the contribution of E6AP R417 to full-length p53 ubiquitination, we generated mutant recombinant E6AP R417A based on the cryoEM structure analysis, which is predicted to block these hydrogen bonding interactions between E6AP and p53[core]. Using carefully measured concentration of E6AP or E6AP R417A we demonstrated that E6AP R417A modestly reduced in vitro p53 ubiquitination by approximately two-fold vs. wild type E6AP at 30 min., whilst autoubiquitination of E6AP R417A was similar to that of parental E6AP (Fig. 6c, d, Supplementary Fig. 26). We then designed R417A/R418A and R417E/R418E double mutants based on aforementioned MD simulation results and analyzed them in ubiquitination assays. Compared with the single mutants (Supplementary Fig. 26), the R417A/R418A double mutant and the double charge reversal mutant, R417E/R418E exhibited a significant reduction in p53 ubiquitination levels, whilst E6AP auto-ubiquitination levels remained quantitatively

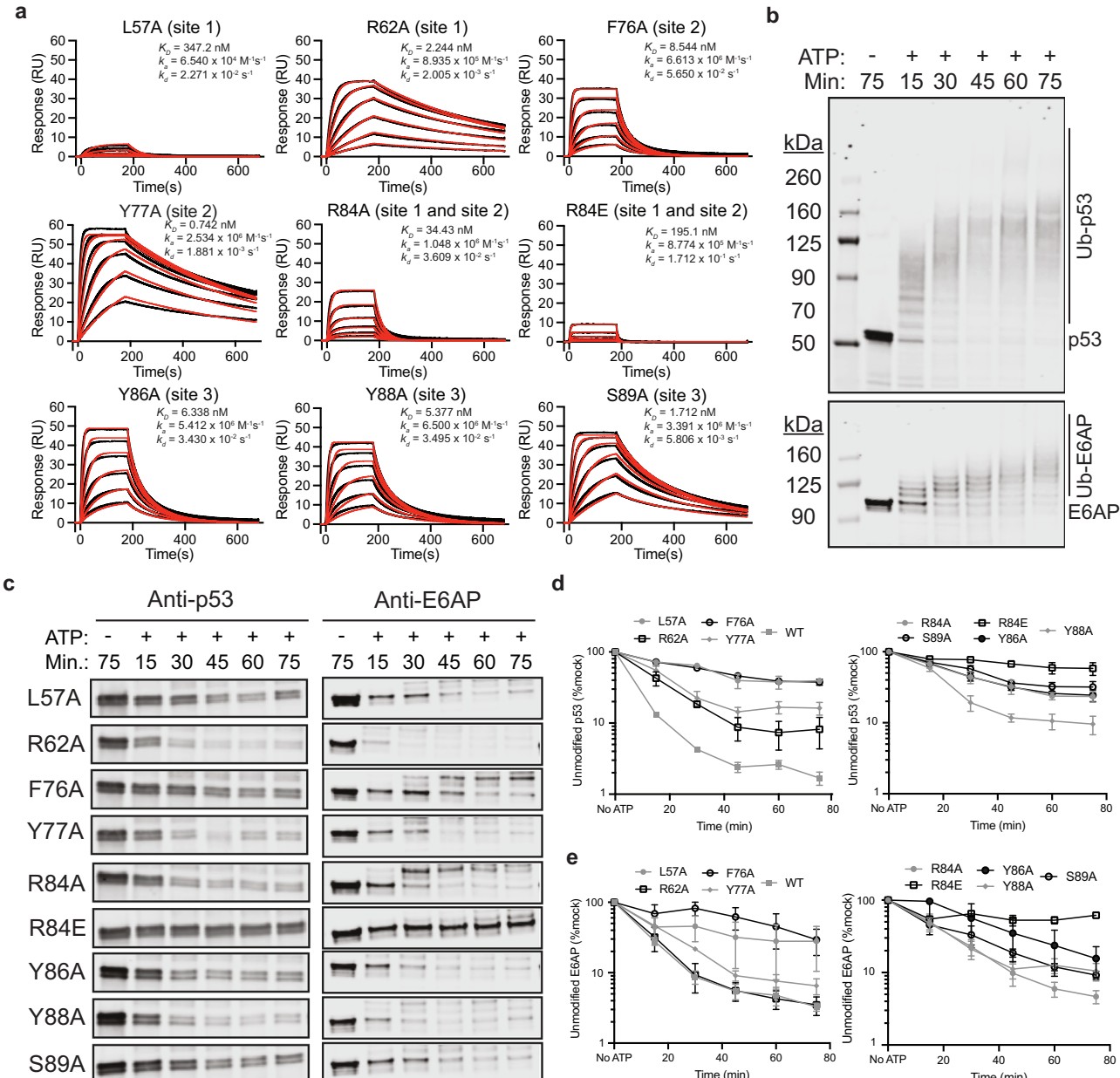

**Fig. 5 | Rationally designed MBP-16E6 mutations disrupt its binding to E6AP. a** Surface plasmon resonance (SPR) measurements of increasing concentrations of mutant MBP-16E6 binding to biotinylated E6AP. Mutations were rationally designed to disrupt the 16E6 and E6AP interaction, reducing the picomolar interaction observed in wild-type protein. The Figure shows measured binding responses (black) and curve fits to a 1:1 interaction model (red). Plots are representative from at least three independent experiments with similar results. RU, response units; $K_D$, dissociation constant; $k_a$, association rate; $k_d$, dissociation rate. **b** p53 ubiquitination assays wherein recombinant MBP-16E6, E6AP, and p53 were incubated in the presence or absence of ATP over the indicated time, then resolved by SDS-PAGE

and visualized for p53 (top panel) or E6AP (bottom panel) by Western blot. **c** Assay was set up as in (**b**) except wild-type MBP-16E6 was substituted with mutant protein and the unmodified p53 (left panels) or unmodified E6AP (right panels) were isolated. Reduced signal represents modification of p53 or E6AP by ubiquitination. Full blots with molecular weight markers can be found in Supplementary Fig. 20. **d** Quantification of unmodified p53 or (**e**) unmodified E6AP from experiments in (**c**). Values represent the mean ± SEM of three independent experiments for all mutant MBP-16E6 (4C4S) proteins, and six independent experiments for WT MBP-16E6 (4C4S) protein.

unchanged (Fig. 6c, d, Supplementary Figs. 21, 26). At the 75 min. time-point, only 1.3% of unmodified p53 substrate remained for WT E6AP; whereas for R417E / R418E, 25.4% remained; and qualitative differences in p53 ubiquitination processivity could also be observed. Conversely, at the 75 min. time-point, 2.22% of unmodified E6AP substrate remained for WT E6AP, and for R417E/R418E E6AP this value was 3.86% (Fig. 6c, d, Supplementary Figs. 21 and 26). These data therefore suggest that E6AP R417/R418 play pivotal roles in direct p53 interaction upon the binding with 16E6. Collectively,

these findings refine the interaction interface between the 16E6:E6AP complex and p53, demonstrating that rational mutations designed to block p53 recruitment can decouple p53 ubiquitination from the ability of 16E6 to promote E6AP autoubiquitination.

### HPV18 E6 exists in higher molecular weight species

The extensive interaction interface and strong binding affinity between 16E6 and E6AP indicate that chemical blockade of the heterodimeric 16E6 and E6AP complex poses a significant challenge. To

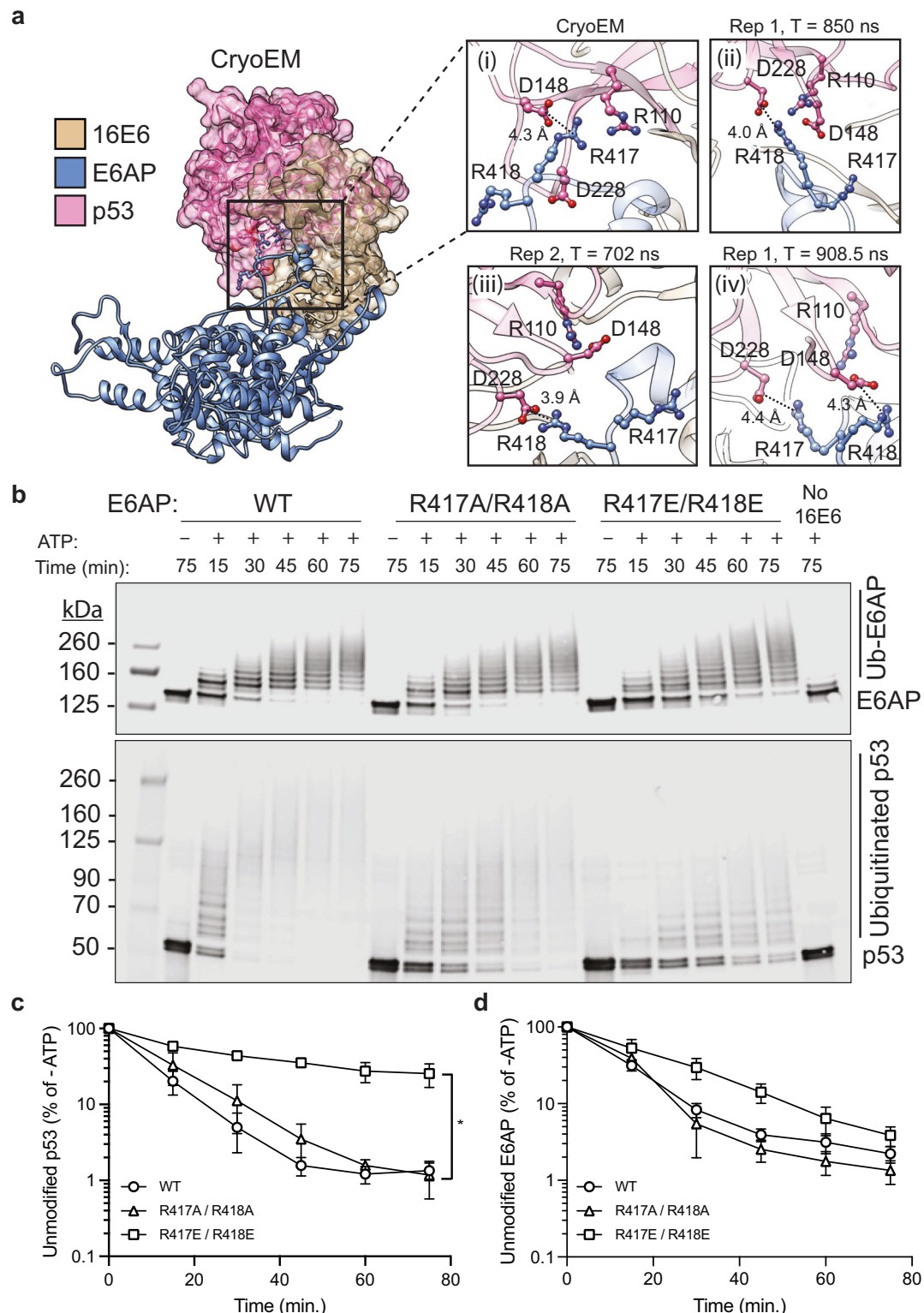

**Fig. 6 | Molecular dynamics simulations identified E6AP R417/R418 as a putative p53 binding residues. a** Structure of the cryoEM ternary complex formed between 16E6 (tan), E6AP (blue), and p53$^{core}$ (purple), with (i) a zoomed-in insert highlighting interactions between E6AP R417 and p53$^{core}$ D148 and R110. (ii-iv) Dynamic and interchangeable interactions between E6AP residues R417 and R418 and p53, as revealed by MD simulations, coordinate the direct interface between E6AP and p53. Hydrogen bonding and ionic interactions are highlighted (black, dashed lines). **b** Results of ubiquitination assays where MPB-16E6(4C4S), WT or

double mutant E6AP, and full-length p53 were incubated in the presence or absence of ATP, resolved by SDS-PAGE, and visualized by western blot for E6AP (upper blot) or p53 (lower blot). Quantification of (**b**) for unmodified (**c**) p53 and (**d**) E6AP. In each (**c**) and (**d**) WT E6AP (black, open circles), R417A/R418A (black, open triangles), and R417E/R418E (black, open squares). Values represent the mean ± SEM of three independent experiments. $P$ * = 0.0002 for unmodified p53 levels at 75 min. between R417E/R418E and WT E6AP by one-way ANOVA.

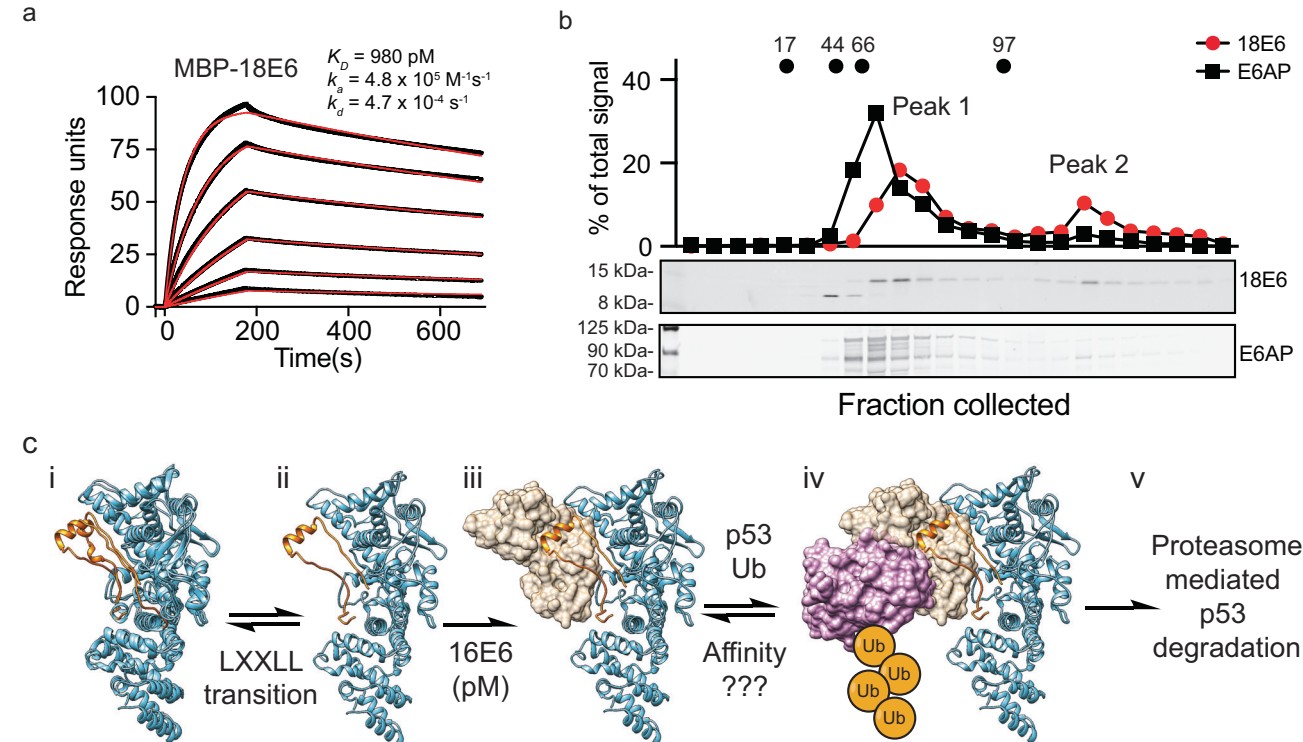

**Fig. 7 | Sedimentation gradient of HPV18+ HeLa cells indicate that 18E6 is fully occupied at endogenous levels. a** SPR measurements of MPB-18E6 binding to biotin-E6AP at varying concentrations are shown. The black line represents the mean binding response of three independent experiments, while the red line represents the curve fit to a 1:1 interaction model. $K_D$, dissociation constant; $k_a$, association rate, $k_d$, dissociation rate. **b** HeLa lysate was fractionated by 20–60% sucrose gradient at 367,600 g for 24 h at 4 °C, and fractions analyzed by SDS-PAGE and blotted for 18E6 (upper blot) or E6AP (lower blot). The quantification of 18E6 (red circles) or E6AP (black squares), with values representing the mean of two independent experiments, is shown above the blots. The peak fractions for fluorescent Myoglobin at 17 kDa, Ovalbumin at 44 kDa, BSA at 66 kDa, and Phosphorylase B at 97 kDa are also shown above. **c** Schematic diagram illustrates the 16E6 and E6AP-mediated degradation of p53. After 16E6 integration into the host genome, 16E6 is expressed and recruited by E6AP through its picomolar binding affinity. (i-ii) The AlphaFold2 E6AP model suggests the flexibility of the E6AP LXXLL motif, and is depicted as a transition state. (iii) After 16E6:E6AP complex formation, (iv) p53 is recruited, ubiquitinated (Ub, orange circles), and (v) degraded by the proteasome which promotes cell survival.

assess the accessibility of the HPV E6 and E6AP interaction interface we sought to quantify the level of unbound 16E6 in a cellular environment. Commercially available antibodies targeting 16E6 have low sensitivity and specificity, with the best antibodies tested having a prominent off-target band immediately below the putative 16E6 band (Supplementary Fig. 27a), while those targeting 18E6 and E6AP have sufficient signal and on-target specificity for western blot analysis (Supplementary Fig. 27b-c). Therefore, we chose to use the HPV18+ HeLa cell line as a surrogate for HPV16+ cells which has 55.6% sequence identity with 16E6 (Supplementary Fig. 1). To ensure that 18E6 interacts with E6AP similarly, we purified recombinant MBP-18E6 (Supplementary Fig. 28) and measured its binding to biotinylated E6AP by SPR with a $K_D$ of 980 pM (Fig. 7a) and its functionality in p53 ubiquitination assays (Supplementary Fig. 29).

To monitor free 18E6 protein in a cellular environment, we fractionated HeLa lysates by ultracentrifugation to monitor 18E6 and E6AP coelution profiles. In an attempt to evaluate disruption of this complex we knocked down 18E6 or E6AP by siRNA in HeLa cells (Supplementary Fig. 27c-g); however, E6AP knockdown reduced 18E6 protein levels to 51% of wild-type and conversely, 18E6 knockdown increased E6AP levels to 177% of wild-type, consistent with previous reports[38]. Due to this feedback loop, 18E6 levels were too low for confident detection by western blotting, therefore we used wild-type HeLa cells for subsequent analysis. A previous study examined 16E6 binding to E6AP in cell lysates by size exclusion chromatography, but artifacts were introduced due to over-expression of HA-tagged 16E6 in SiHa cells[39]. To overcome previous confounding issues, we lysed HPV18+ HeLa cells and used sucrose gradient sedimentation to separate protein

complexes based on their density. The sucrose gradient fractions were separated by SDS-PAGE, and western blot analysis of 18E6 and E6AP proteins was performed.

In all experiments we included fluorescently labeled molecular weight standards and incorporated them into our workflow to ensure inter-experimental compatibility (Supplementary Fig. 27h-j). Surprisingly, we found that 18E6 eluted in two distinct high molecular weight fractions containing E6AP, with the major peak (peak 1) eluting between BSA (66 kDa) and phosphorylase B (97 kDa) despite the predicted monomeric molecular weight of 18 kDa (Fig. 7b). The second peak (peak 2) fractionated at a molecular weight larger than 97 kDa (Fig. 7b). Peak 1 and peak 2 are consistent with previous SEC-based overexpression experiments; however, in that study a third low molecular weight fraction containing monomer/dimer HA-16E6 was also observed, in addition to an HA-16E6:E6AP peak at approximately 150–200 kDa and larger complex containing HERC2, E6AP, and HA-16E6[39]. However, it is a formal possibility that low molecular weight 18E6 may exist outside western blot detection limit. Additionally, we found that the absence of HPV E6 in cells had no effect on E6AP distribution across the sucrose gradient as E6AP distribution was similar between HPV18+ HeLa and HPV negative HT1080 cells (Supplementary Fig. 27k-l) consistent with previous studies[39].

This observation that 18E6 is occupied in higher molecular weight species led us to question the relative abundance of 18E6 and E6AP at the protein level in the cellular environment. To ascertain this in a semi-quantitative manner, we probed increasing amounts of HT1080 or HeLa lysate for 18E6 (in the case of HeLa) and E6AP (for each) (Supplementary Fig. 30). On the same blot we ran increasing amounts

of recombinant MBP-18E6 or E6AP to generate standard curves to interpolate the amount of protein present in the lysate. Our data suggest that the amount of 18E6 in HeLa cells (~2.9 femtomoles/μg of lysate) is greater than that of E6AP (~1.0 femtomoles/μg of lysate) indicating a likely molar excess of 18E6 compared with E6AP. Taken together, our data suggest that in HeLa cells, the majority of 18E6 is occupied within higher molecular weight complexes although it is unclear what percentage of 18E6 is occupied by E6AP, as opposed to its other known binding partners[14,40].

## Discussion

HPV is a prevalent sexually transmitted infection associated with various cancers, including cervical, head and neck squamous cell carcinoma, and anal cancer. Although several treatment options exist, they often come with long-term side effects. Immunotherapies targeting E6 and E7 antigens have shown promise in treating HPV-associated cancers, but targeted therapeutics have gained limited traction, possibly due to the lack of structural understanding of this complex and endogenous behavior of 16E6. To address this, we solved the cryoEM structure of the 16E6, E6AP, and p53[core] ternary complex, revealing an extensive and high-affinity interaction interface and revealed that endogenous unbound 18E6 is not found in lysates resolved by sucrose gradient sedimentation.

Our findings demonstrate that 16E6 binds to E6AP with picomolar binding affinity through a large 2361 Å² interface, providing a structural rationale for this unexpectedly high affinity (Fig. 1a). Recent yeast two hybrid studies implicated alternative 16E6 interaction sites with E6AP residues. They show that the first 127 E6AP residues contribute to the association of 16E6 L57A with E6AP, but were not required for 16E6 stimulation of E6AP ubiquitin ligase activity[30], although this region was not resolved in our structure (Supplementary Fig. 11). Additionally, truncation of E6AP residues 310–320 were identified as important for p53 degradation although no single point mutation that reproduced this phenotype were identified[30]. Although E6AP residue L318 is within ~6 Å of 16E6 E98, no direct interactions are observed in the cryoEM structure. We identified three interaction sites between 16E6 and E6AP that mediate dimeric complex formation and that are required for p53 ubiquitination, verified through rational mutagenesis (Fig. 5). Next, we demonstrated that it is possible to uncouple E6AP and 16E6 complex formation from subsequent p53 ubiquitination. Introduction of a double mutation charge reversal, R417E and R418E, significantly reduced p53 ubiquitination levels, whilst having modest effects on E6AP auto-ubiquitination levels (Fig. 6). This is an important finding as it suggests alternative druggable sites for disrupting p53 ubiquitination, restoring p53 status, and resulting HPV-dependent cell death.

Comparative analysis using MD simulations of the cryoEM E6AP and the AlphaFold2 model in their apo- and complex-states, revealed that the LXXLL motif is highly flexible in the apo-state. However, it seems likely that 16E6 can recognize and further stabilize the E6AP LXXLL motif in a specific confirmation supported by both MD simulation and structural analysis (Figs. 2−4), suggesting that 16E6 binds to E6AP with high specificity. Although the consequence of limiting the dynamics of the LXXLL motif is unclear in terms of E6AP substrates, the AlphaFold2 structure is incompatible with 16E6 binding. Outstanding questions remain surrounding E6AP HECT domain organization and p53 recruitment, which will require additional studies including methodical structure-function analyses and MD simulations.

The pharmaceutical industry's enthusiasm for developing therapeutics targeting HPV E6 has decreased since the introduction of HPV-directed vaccines over ten years ago, due to the perception of weak commercial opportunities. However, HPV+ cancers remain a significant unmet medical need due to the lack of widespread distribution, uptake, availability, and a large population of unvaccinated individuals outside of administration guidelines[41,42]. Multiple efforts have been made to identify E6-directed therapeutics, including siRNAs[43,44], chemical matter, including affibodies, nanobodies[45,46], intrabodies[47,48], and small molecule inhibitors[49]. Even further efforts have been made to target p53[50] or negative regulators of p53[51], yet no p53 drug discovery initiative has received FDA or EMA approval[52,53]. The 16E6 LXXLL-binding groove was thought to be an attractive target for small molecule drug discovery due to its hydrophobic binding groove which appears to have many favorable features for targeted disruption via chemical perturbation. Previous studies have targeted this groove using polyhydroxy flavonoids[54,55], LXXLL-derived mini proteins[56], and LXXLL-based peptides[35,37,57–59]. However, targeting site 1 is a challenging proposition, as demonstrated by this work, and the lack of translatability may originate from the liabilities described above, including the picomolar binding affinity between 16/18E6 and E6AP, and the possibility that free 18E6 is unavailable in HPV+ cells. Nonetheless, our finding that specific residues on E6AP mediate p53 ubiquitination, whilst E6AP is bound to HPV E6, may provide unrealized opportunities for on-going therapeutic efforts to decouple p53 ubiquitination from E6 and E6AP recognition. Our cryoEM structure and MD-simulation, together with comprehensive functional analysis, provide insights into potential therapeutic strategies.

## Methods

### Generation of DNA constructs

The full-length E6AP (Uniprot ID no. Q05086) protein was C-terminally fused to a TEV-6xHis-Avi sequence, and was sub-cloned into a pFast-Bac1 vector (Genscript). The MBP-GGGGS-TEV sequence was fused to the N-terminus of full-length 16E6 (Uniprot ID no. P03126), and the nucleotide sequence was inserted into a pET28a vector (Genscript). Four point mutations, C87S, C104S, C118S, and C147S, were introduced into the 16E6 construct for cryoEM study (Plasmids synthesized by Genscript). The coding sequence of p53[core] domain (Uniprot ID no. P04637, S94-T312) fused with an N-terminal 8xHis-lipoyl domain (Uniprot ID no. P11961, A2-T85)-TEV sequence, was sub-cloned into a pET21a vector (Genscript). All constructs were codon-optimized (GenSmart™ Codon Optimization, Version Beta 1.0) against their respective expression hosts and synthesized by Genscript.

### Protein expression and purification

BL21(DE3) cells (Thermofisher, cat. no. EC0114) were transformed with plasmids containing truncated p53 and HPV E6 mutant coding sequences. The cells were grown in LB media supplemented with ampicillin (50 μg/mL) or kanamycin (50 μg/mL) at 37 °C at 150 rpm to an optical density of 0.8 measured at 600 nm using a UV spectrophotometer (Unicosh, UV-2800A). Protein expression was induced with 0.5 mM IPTG (Sigma, cat. no. 16758) for 16 h at 15 °C while shaking at 200 rpm. Bacmids and recombinant viruses of E6AP were prepared according to the Thermo Fisher instructions (Invitrogen Version A, A10606) and were amplified in Spodoptera frugiperda Sf9 cells (Thermo Fisher, cat. no. 11496-015). P2 viruses were used at 2 μL/mL of virus/media for protein expression in Sf21 cells, and cells were harvested 48 h post-infection. Primary amino acid sequence of MPB-16E6 protein is found in Supplementary Fig. 3a.

Protein purification was performed at 4 °C unless otherwise stated. BL21(DE3) cells expressing the p53[core] domain were collected by centrifugation at 3470 g for 10 min., and the cell pellets were resuspended in lysis buffer containing 50 mM Tris-HCl (pH 7.5), 300 mM NaCl, 5 mM DTT, 10% glycerol, 5 mM imidazole, 1 mM PMSF, 1 mM benzamidine, and cOmplete Protease Inhibitor (Roche; cat. no. 53002800). Cell lysates were clarified through centrifugation at 38,900 g for 30 min., the supernatant was loaded onto a His Excel column (cat. no. 17371202) pre-equilibrated in 50 mM Tris-HCl pH 7.5, 300 mM NaCl, 5 mM DTT, 10% glycerol, 5 mM imidazole, 1 mM PMSF, 1 mM benzamidine wash buffer. Proteins were washed with 10 CVs and eluted in the same buffer supplemented with 500 mM imidazole. The protein elution was incubated with TEV protease (made by Biortus Co. LTD) overnight and the tag was

further removed by the His Excel column. The non-tagged p53 protein was subjected to size exclusion chromatography using a HiLoad 16/600 Superdex 75 pg (cat no. 28989333) pre-equilibrated in 50 mM Tris-HCl pH 7.0, 150 mM NaCl and 5 mM DTT. Protein fractions were pooled based on SDS-PAGE gel stained with Coomassie assessment and concentrated to approximately 2 mg/mL by Amicon® Ultra-15 (Millipore). The final concentration was determined by absorbance at 280 nm using Nanodrop (Merinton, SMA5000), flash-frozen in liquid nitrogen and stored at −80 °C.

*E. coli* cells expressing HPV E6 were disrupted by the French press (Union-Biotech, UH-06), and cell lysate was cleared by centrifugation at 38,900 $g$ for 30 min. Supernatant was passed over an MBPTrap HP column (cat no. 28935597) equilibrated in 50 mM Tris-HCl, pH 7.0, 400 mM NaCl, 5 mM DTT. Protein was eluted with the same buffer supplemented with 25 mM maltose (Sigma, cat. No. 63423), and subjected to size exclusion chromatography on a HiLoad 16/600 Superdex 200 pg column (cat no. 28989335) in 50 mM Tris-HCl, pH 7.0, 150 mM NaCl, 5 mM DTT. Fractions were pooled, concentrated using Amicon® Ultra-15 (Millipore) to ~3.9 mg/mL determined by absorbance measured at 280 nm using Nanodrop (Merinton, SMA5000). Proteins were flash frozen in liquid nitrogen and stored at −80 °C.

*Sf21* cells expressing E6AP were lysed by French press in 50 mM HEPES, pH 7.5, 500 mM NaCl, 5 mM imidazole, 5% glycerol, 1 mM PMSF, and cOmplete Protease Inhibitor (Roche; cat no. 53002800). Lysate was clarified by centrifugation for 30 min. at 39,800 $g$, E6AP was enriched by Ni Bestarose FF resin (Bestchrom, cat no. AA0053), then washed with 10 CVs of 50 mM HEPES, pH 7.5, 500 mM NaCl, 5% glycerol, 1 mM PMSF and 20 mM imidazole, then eluted with the same buffer supplemented with 500 mM imidazole from the Ni Bestarose FF resin (Bestchrom, cat no. AA0053). Primary amino acid sequence of E6AP protein is found in Supplementary Fig. 6a. The eluted fraction was diluted and subjected to a Mono Q 10/100 GL column (Cytiva, cat no. 17516701) for further purification with a 20 CV linear gradient elution (Buffer A: 50 mM Tris-HCl (pH 7.5), 100 mM NaCl, 5% glycerol, 1 mM PMSF; Buffer B: 50 mM Tris-HCl (pH 7.5), 1 M NaCl, 5% glycerol, 1 mM PMSF). Peak fractions were combined based on Coomassie staining of enriched fractions after resolving by SDS-PAGE. Peak fractions were then concentrated on Amicon Centrifugal Filters (cat no. R1SB42368), and further purified by size exclusion chromatography on a HiLoad 16/600 Superdex 200 pg column (Fisher Scientific, cat no. 28989335) equilibrated in 25 mM HEPES pH 7.5, 150 mM NaCl, 1 mM TCEP. Proteins in the peak absorbance at 280 nm fractions of the SEC profile were pooled and concentrated to 5.9 mg/mL by Amicon Centrifugal Filters (Fisher Scientific, cat no. R1SB42368), flash frozen in liquid nitrogen and stored at −80 °C. All purification steps were done on FPLC (GE Healthcare, AKTA FPLC) using UNICORN version 7.10 control software.

## Analytical HPLC of 16E6 mutants

MBP-tagged HPV16 E6 mutants were analyzed on a Superdex200 Increase 5/150 GL column (Cytiva, cat no. 28990945) using an Agilent 1200 Series Infinity II HPLC equilibrated in PBS, 1 mM DTT at room temperature using OpenLab CDS version C.01.10. Protein samples were diluted to 2 mg/mL using the equilibration buffer and 20 µL were injected over the column and ran isocratically at 0.4 mL/min. over 8 min. A standard curve to relate elution time to molecular weight was generated using an injection of 15 µL of Gel Filtration Standard (BioRad, cat no. 1511901) diluted to 1:10 of manufacturer's recommended concentration following the same method.

## CryoEM sample preparation and data collection

The complex for the cryoEM structural determination was prepared in two stages. Individually purified MBP-16E6 and E6AP proteins were mixed then incubated on ice for 1 h, at a 2:1 molar ratio, respectively. Samples were then concentrated by Amicon Centrifugal Filters (Fisher Scientific, cat no. R1SB42368) and resolved using a Superdex 200

Increase 10/300GL column (Cytiva, cat no. 28990944) equilibrated with 50 mM Tris-HCl (pH 7.0), 150 mM NaCl, 5 mM DTT, 0.01% Chapso (Sigma-Aldrich, C3649). Peak fractions of the SEC profile as determined by absorbance at 280 nm were pooled and concentrated using Amicon Centrifugal Filters (Fisher Scientific, cat no. R1SB42368) to a concentration of approximately 1.5 mg/mL as determined by absorbance at 280 nm using Nanodrop (Merinton, SMA5000). Preparation of the 16E6:E6AP:p53$^{core}$ trimeric complex was accomplished by incubating the 16E6:E6AP sub-complex with tag-removed p53$^{core}$ domain on ice for 1 h, at a 1:2.5 molar ratio. The 16E6, E6AP, and p53$^{core}$ ternary complex was then resolved on a Superdex 200 Increase 10/300GL column pre-equilibrated in 50 mM Tris-HCl pH 7.0, 150 mM NaCl, 5 mM DTT, 0.01% Chapso. Fractions corresponding to the 16E6:E6AP: p53$^{core}$ ternary complex were collected and concentrated using Amicon Centrifugal Filters (Fisher Scientific, cat no. R1SB42368) to approximately 2.6 mg/mL as determined by absorbance at 280 nm using Nanodrop (Merinton, SMA5000). The complex was flashed frozen in liquid nitrogen and stored at −80 °C for further usage. The sample was subjected to centrifugation (20,600 $g$ for 10 min.) after freeze and thaw to get rid of potential precipitation before grid preparation.

To prepare grids for data collection, 3 µL of 16E6, E6AP and p53$^{core}$ ternary complex (2.15 mg/mL) was applied to a 300 mesh R 1.2/1.3 holey carbon Film (Quantifoil; lot no. 230634) grid which was glow-discharged for 60 s at 20 mA using a glow-discharge cleaning system (PELCO easiGlow; cat. no. 91000-00492). After incubating at 4 °C and 100% relative humidity for 1 min., excess solution was removed by blotting for 6 s with a blotting force of 4. The grid was then immediately plunge-frozen in liquid ethane using a Vitrobot Mark IV system (Thermo Fisher Scientific; cat. no. 190100729), and stored in liquid nitrogen prior to data acquisition.

CryoEM movies were collected on a Titan Krios transmission electron microscope (Thermo Fisher Scientific) operated at 300 kV. Images were collected with a K3 direct electron detector operating in super-resolution counting mode for recording data, with the slit width of the Gatan Quantum GIF (Gatan, USA) set to be 20 eV. Data collection was automated using SerialEM[60] at a nominal magnification of 105 K, resulting in a physical pixel size of 0.83 Å. A total dose of 80.5 e⁻/Å² was fractionated into 50 frames. In total, 8127 movies were collected with a defocus range of −1.5 to −2.5 µm.

## Image processing, 3D reconstruction

Movies were aligned and dose-weighted using MotionCor2[61] through RELION version 3.0.8.[62] Dose-weighted summed images were imported into cryoSPARC (v3.0.1) for further processing. The contrast transfer function (CTF) parameters were estimated using Patch CTF, and bad images were removed through visual inspection and with a cutoff value of 4.5 Å resolution, as estimated by Patch CTF in CryoSPARC[63]. After curation, 6761 good images (high resolution, complete and clean hole, clear particles in the hole) were preserved. Blob picking was employed on a small subset (~500 images) of images to pick particles, followed by 2D classification to generate templates for later processing. A set of 2,349,301 particles were picked by template picking, and were subjected to three rounds of 2D classification clean-up, which gave rise to a subset of 1,605,377 particles. The particle subset went through ab-initio reconstruction and several rounds of heterogeneous refinement, resulting in a smaller subset containing 233,383 particles. A final round of ab-initio and heterogeneous refinement were executed to generate a final set of 105,794 particles, which were then subjected to non-uniform refinement. To further improve the map quality, the density of the HECT domain (E6AP residues from 776 to 875) was erased and the resultant map was used as the input for a final step of non-uniform refinement in cryoSPARC[63] due to the weak density observed (Supplementary Fig. 13d). The resolution of final reconstruction was estimated to be 3.38 Å based on the gold-standard Fourier shell

correlation using the 0.143 criterion[64]. A global B-factor of −113 Å² was applied to sharpen the density map. Local resolution estimation was done using Relion's own implementation[62].

### Model building

To build the 16E6, E6AP, and p53core ternary complex, the crystal structures of 16E6 and p53core from PDB ID no. 4XR8, were rigid body fitted into the electron density map, followed by fitting the AlphaFold2-predicted model of E6AP[65] as a rigid body in Chimera[66]. Subsequently, we performed several rounds of iterative manual rebuilding/adjustment in COOT[67], followed by refinement using Refmac5 in the CCPEM package[68] to improve the model quality. We evaluated the final model using MolProbity[69]. CryoEM data statistics are summarized in Supplementary Table 1.

### Protein structural analysis

Manuscript was written in Microsoft Office (version no. 16.56) and Figures were constructed in Adobe Illustrator (version 26.1). Protein structures were depicted using either UCSF Chimera version 1.14[66], UCSF ChimeraX version 1.4[70], PyMOL (Schrödinger, LLC, version 2.5.2) or MOE (Chemical Computing Group, version 2020.09.01). "Molecular graphics and analyses performed with UCSF Chimera, developed by the Resource for Biocomputing, Visualization, and Informatics at the University of California, San Francisco, with support from NIH P41-GM103311." "Molecular graphics and analyses performed with UCSF ChimeraX, developed by the Resource for Biocomputing, Visualization, and Informatics at the University of California, San Francisco, with support from National Institutes of Health R01-GM129325 and the Office of Cyber Infrastructure and Computational Biology, National Institute of Allergy and Infectious Diseases." MOE is cited as followed: "Molecular Operating Environment (MOE), 2020.09; Chemical Computing Group ULC, 1010 Sherbrooke St. West, Suite #910, Montreal, QC, Canada, H3A 2R7, 2020."

The root mean square deviations (RMSDs) were calculated using the α-carbon of each aligned and superimposed residue of the indicated structures using the MatchMaker utility within the UCSF Chimera suite. To calculate protein:protein binding energies and to define protein-protein interactions the finalized 16E6, E6AP, and p53core complex was imported into MOE (Chemical Computing Group, version 2020.09.01). The structure was solvated and protonated using the protonate 3D utility with the default setting in the Amber10:EHT forcefield. Where indicated, the protein interaction interfaces were isolated, and the binding energy calculated (kcal/mol; GBVI) using the protein contact utility which describes the sum of all interactions at the interface. For individual protein contacts the protein contact utility in MOE was used to define interactions using the default settings and a summary of all interactions found in Supplementary Data 1. The protonated 16E6, E6AP, and p53core ternary complex was exported in the MOE file format and can be found as Supplementary Data 2 and the exported PDB file format is found as Supplementary Data 3. For files AF-Q05086-F1-model_v1 and AF-Q05086-F1-model_v3, the version numbers were different due download date although the rmsd between the two structure files is zero. To maintain transparency, the file names were maintained.

### Molecular dynamics simulations

Prior to performing the MD simulations on the cryoEM ternary complex of E6AP, 16E6, and p53core we introduced missing regions in E6AP using the AlphaFold2[65] predicted structure for guidance. Specifically, we added in the short missing regions of 359R-N363, 386G-D389, and G433-R440 of E6AP that were not resolved by cryoEM. We then superimposed our cryoEM structure onto the predicted AlphaFold2 structure using the Needleman-Wunsch alignment algorithm[71] with the BLOSUM-62 matrix, which was integrated in UCSF Chimera software[66].

Next, we further refined and prepared the improved construct using Molecular Operating Environment (MOE), 2022.02 Chemical Computing Group ULC, 910-1010 Sherbrooke St. W., Montreal, QC

H3A 2R7, Canada, 2024 QuickPrep module. This step involved adjusting the protonation states of all atoms and minimizing the structure of the cryoEM system using the Amber10:EHT force field, which was incorporated in the MOE-2022.02 software package. During the minimization process, we applied MOE default tether restraints to the protein atoms and ensured that the root-mean-square (RMS) gradient of the potential energy remained below 0.1 kcal mol$^{-1}$ Å$^{-2}$.

We chose not to model in the large missing region between V164-V234 to prevent any unnecessary artifacts that this region could introduce on the behavior of the entire ternary complex. Instead, we used the neutral terminal carboxylate of V164 as COOH and the neutral form of the amine group of V234 as $NH_2$. Then, we placed a lower and upper wall constraint on the distance between the Cα atoms of V164 and V234 to be between 8.4 Å and 10.0 Å, with a force constant of 12.0 kcal.mol$^{-1}$Å$^{-2}$. This constraint allowed the two isolated regions to freely fluctuate within the restricted space without losing their link with the entire E6AP protein body. We implemented these constraints using the PLUMED-2.8.0 software package[72]. Finally, we used the refined model of the p53core, E6AP and 16E6 ternary complex for all five MD simulation experiments in this study, as described in Supplementary Table 2.

To prepare the protein systems for MD simulations, we first solvated each of the 5 systems in a water box with dimensions specified in Supplementary Table 2. The system was then neutralized by adding excess NaCl to achieve a physiological salt concentration of 150 mM. Next, we performed energy minimization of the system using 1000 steps of the steepest descents algorithm, as implemented in GROMACS software[73]. The minimized system was then heated gradually from 0 K to 310 K in 600 ps during an MD simulation in a canonical ensemble. After that, an MD simulation in an isobaric-isothermal ensemble was carried out for 22 ns to relax the simulation box and equilibrate the system at a pressure of 1 bar. Throughout the pre-equilibration steps, we placed positional restraints on all heavy atoms (non-H atoms) using 47.8 kcal.mol$^{-1}$Å$^{-2}$, which were progressively reduced to 0 kcal.mol$^{-1}$Å$^{-2}$ for the final equilibration step. Finally, we performed MD simulations in an isobaric-isothermal ensemble to equilibrate each of the 5 systems for trajectory analysis.

The Charmm36m parameter set[74] was used to describe protein, and ions, while water was described using the CHARMM TIP3P model. The temperature was maintained at 310 K using a velocity-rescale[75] thermostat with a damping constant of 1.0 ps for temperature coupling and the pressure was controlled at 1 bar using a Parrinello-Rahman barostat algorithm with a 5.0 ps damping constant for the pressure coupling. Isotropic pressure coupling was used during this calculation. The Lennard-Jones cutoff radius was 12 Å, where the interaction was smoothly shifted to 0 after 10 Å. Periodic boundary conditions were applied to all three directions. The Particle Mesh Ewald algorithm[76] with a real cutoff radius of 12 Å and a grid spacing of 1.2 Å was used to calculate the long-range coulombic interactions. A compressibility of $4.5 \times 10^{-5}$ bar$^{-1}$ was used to relax the box volume. In all the above simulations, water OH bonds were constrained by the SETTLE algorithm[77]. The remaining H-bonds were constrained using the P-LINCS algorithm[78]. All MD simulations were carried out using GROMACS-2021[73]. For the results presented in Fig. 4 we used gromos algorithm[79] with RMSD cutoff = 1.0 Å to the entire MD trajectory of E6AP. The molecular dynamics reporting document is included as Supplementary Table 3.

### p53 ubiquitination assay

p53 ubiquitination assays were carried out in 12.5 μL reactions consisting of 100 nM UbE1 (cat. no. E-304, R&D Systems, lot no. DBFR0722091), 200 nM UbE2D2 (cat. no. E2-622, R&D Systems, lot no. 04101722C), 200 nM MBP-16E6 (wild-type 4C4S or mutant 4C4S, as indicated), 200 nM Avi-E6AP (wild-type or mutant), 200 nM full-length p53 (cat. no. 81701, Active Motif, Inc., lot no. 03922002),

0.1 mg/mL ubiquitin (cat. no. U-100H, R&D Systems, lot no. DBDZ2822041), and 10 mM Mg-ATP in 1xE3 ligase buffer (cat. no. B-71, R&D Systems, lot no. 14656722) at 37 °C for the indicated time. Reactions were stopped by the addition of 4x sample buffer (cat. no. 928-40004, LI-COR Biosciences, lot no. D20511-06) supplemented with 10% β-mercaptoethanol (cat. no. M3148, Sigma Aldrich) and heating at 99 °C for 5 min. in a Corning LSE Digital Dry Bath.

Reaction products (4 μL) were resolved by SDS-PAGE (4–20% TGX, cat. no. 5671095) in 1X Tris-Glycine-SDS running buffer, transferred to 0.2 μm nitrocellulose (cat. no. 1704159, BioRad) using the "Mixed Mw Turbo" mode on the Trans-Blot Turbo Transfer System (cat. no. 1704150, BioRad). Nitrocellulose was blocked with Intercept (TBS) blocking buffer (cat. no. 9276001, LI-COR) before being incubated with Intercept (T20, TBS) Antibody Diluent buffer (cat. no. 92765001, LI-COR) supplemented with α-p53 antibody (cat. no. MAB1355, R&D Systems, lot no. IAH0422071) overnight at 4 °C with shaking. Blots were washed with TBS-T, and subsequently incubated with goat-anti-mouse (800 nm conjugate; cat. no. 926-32210, lot no. D11116-25, LI-COR) secondary antibody for 2 h at room temperature with shaking, then washed with TBS-T buffer four times with shaking. To analyze E6AP levels, conditions were the same as for p53, however the anti-E6AP antibody was utilized (Cell Signaling, cat. no. D10D3, lot no. 7526S) followed by incubation with goat-α-rabbit (680 nm conjugate: cat. no. 926-68071 lot no. D21207-05, LI-COR).

To ensure that reactions were quantifiable, we assessed the linearity of the anti-p53 antibody (cat. no MAB1355, lot no. IAH0422071) and the anti-E6AP antibody (cat. no. D10D3) across a dilution series from 1–150 ng for each full-length p53, WT E6AP, or R417A E6AP using a 26-well 4-20% TGX gel (cat no. 5671095) using the aforementioned conditions. For full-length p53 linear signal was observed between 1 and 25 ng per well and for E6AP linear signal was observed up to 150 ng per well; thus enabling the robust quantification of the loss of unmodified full-length p53 or E6AP due to ubiquitination in vitro. Blots were imaged with an ODYSSEY CLx LI-COR instrument using Image Studio software (version no. 5.2) and the following scan settings: automated intensity, 169 μm pixel size, and 0.0 mm offset. All p53 ubiquitination assays were repeated a minimum of three times. Gel bands were quantified using auto-exposure, and background subtraction was carried out using the software. The amount of full-length p53 substrate was quantified and normalized for each 16E6 and E6AP condition using its own no-ATP control. The loss of unmodified full-length p53 substrate, due to generation of ubiquitinated product, was plotted against time as a measure of full-length p53 ubiquitination kinetics.

## HPV E6 and E6AP surface plasmon resonance (SPR)
Biotinylated E6AP was freshly prepared using a biotin-protein ligase kit (Avidity, cat no. BirA500) prior to SPR experiments. A total of 50 μL of 100 mM ATP, 100 mM Mg(OAc)$_2$, and 500 μM of d-biotin was added to 500 μL of 1.7 mg/mL E6AP in 25 mM HEPES pH 7.5, 150 mM NaCl, and 1 mM TCEP. The mixture was supplemented with 20 μg of BirA biotin-protein ligase and incubated at 4 °C for 16 h. Excess biotin was removed using a Superdex 200 Increase 10/300 GL column (Cytiva, cat no. 28990944) equilibrated in PBS supplemented with 1 mM DTT, with a flow rate of 0.5 mL/min. on an ÄKTA Pure 25 M.

16E6 SPR data collection was carried out using a Biacore T200 (Cytiva, cat no. 28975001) using Biacore T200 Control Software Version 3.2.1 while 18E6 SPR data was collected using a Biacore 8K+ (Cytiva, cat no. 29344964) using Biacore Insight Control Software Version 5.0.18 equipped with Series S CAP sensor chips (Cytiva, cat no. 28920234), flow cells set at 25 °C, the sample compartment set at 4 °C, and a Running Buffer of PBS with 1 mM DTT and 0.005% v/v Surfactant P20. Flow cells #1 and #2 were conditioned with Biotin CAPture Reagent (Cytiva, cat no. 29423383) at 2 μL/min. for 300 s. Freshly biotinylated E6AP was diluted to 20 μg/mL using Running Buffer and

injected over the flow cell #2 for 80 s at 2 μL/min to obtain ~90 response units (RU) of immobilized ligand. MBP-E6 proteins were diluted to concentration series of [50.0, 25.0, 12.5, 6.25, 3.12, 1.6] nM and injected over both flow cells 1 and 2 for 180 s of association followed by 500 s of dissociation at 30 μL/min. Analyte concentration was limited to a maximum of 50 nM due to non-specific reference binding onto conditioned CAP sensor chips above the chosen concentration (Supplementary Fig. 4a). The chip surface was regenerated using 6 M Guanidine-HCl and 0.25 M NaOH, with a 60-s first injection and a 30-s second injection at 30 μL/min. A final wash step over the chip surface was performed using a 30-s injection of PBS at 30 μL/min. To ensure E6AP did not degrade during the course of experiments as biotinylated E6AP loses binding capability over time, a control injection of 20 nM MBP-HPV16 E6(4C4S) was performed to measure whether the maximum signal still matched the theoretical maximum ligand binding capacity, Rmax, at the end of each set of experiments (Supplementary Fig. 31).

The Biacore T200 Evaluation Software version 3.2.1 was used for the analysis of SPR data. A kinetics 1:1 binding model was applied to fit the curves, with the RI value set to a constant of 0, and the theoretical maximum ligand binding capacity (Rmax) constrained.

## In vitro 16E6 and E6AP NanoBRET assay
Recombinant E6AP-Nluc, MBP-16E6-Halo and its mutants were produced as for their parental constructs described above. Alignment of proteins used in NanoBRET assays compared to their parental constructs used for SPR and ubiquitination assays can be found in Supplementary Fig. 6a. The Halo and Nluc tags, as well as linker region were based on Promega kit guidance; 16E6 NanoBRET vector (cat no. CS1679C142A, Promega); NanoBRET assay donor vector, E6AP (cat no. CS1679C142B, Promega). Analytical HPLC and SDS-PAGE gels of the NanoBRET proteins can be found in Supplementary Fig. 32 while LC-MS characterization data can be found in Supplementary Fig. 33.

To run the reaction, MBP-16E6-Halo (1 μM) was incubated with Halo 618 ligand (2 μM, Promega G9801) at 4 °C for 2 h, and conjugation was confirmed by LC/MS (Supplementary Fig. 6a). DMSO or Halo 618 ligand-labeled MBP-16E6-Halo proteins were serial diluted by two-fold from 1 μM to 1.95 nM with the PBS, 1 mM DTT and 1:1 mixed with 100 pM E6AP-Nluc for 1 h on ice to reach equilibrium. Nano-Glo Luciferase Assay Substrate (Promega N113A) was added to the mixture with a final dilution of 1:200. Protein mixture (20 μL) was transferred to an Opti-plate 384-well, white Opaque 384-well microplate (Perkin Elmer 6007290) and the donor emission (460 nm) and acceptor emission (618 nm) were measured immediately on the ClarioStar plate reader (BMG Labtech) using the MARS software (version 3.42 R4). The filter setup included a band pass (BP) filter centered at 460 nm to measure the donor signal (Emission 450 nm/BP 80 nm) and a long pass (LP) filter at 600–610 nm to measure the acceptor signal (Emission 610 nm/ LP). BRET ratios were calculated following the guideline from the Promega manual (NanoBRET Protein-Protein Interaction System Technical Manual #TM439).

## Antibody validation in lysates
To test the 16E6 antibodies, HT1080 (HPV negative; ATCC, cat. no. CCL-121, lot no. 70032964), HeLa (HPV18; ATCC, cat. no. CCL-2 lot no. 70033477) and CaSki (HPV16; ATCC, cat. no. CRM-CRL-1550, lot no. 70032964) cells were grown to 70–80% confluency and lysed by addition of cold Pierce IP lysis buffer (cat. no. 87787, lot no. WK337252) supplemented with 1X Halt Protease Phosphatase inhibitor cocktail (Thermo, cat. no. 1861284, lot no. XC342080) to the plates for 10 min on ice. Cells acquired from ATCC had undergone STR profiling prior to purchase. During their usage, cells were tested monthly for mycoplasma contamination, consistently yielding negative results. Plates were scraped and spun at max speed in a bench top centrifuge for 10 min. at 4 °C. Supernatant was removed

and lysates quantified using a Pierce 660 assay (Thermo, cat. no. 22660, lot no. XA338566). 20 μg from each cell line was resolved on a 4–20% TGX criterion gel, transferred to nitrocellulose blocked with TBS intercept blocking buffer (LI-COR, cat. no. 927-65001) then incubated overnight at 4 °C with either the HPV16 E6 (Genetex, cat. no. 132686, lot no. 44545) or the HPV18 E6 antibody (Genetex, cat. no. GTX132687, lot no. 42592) antibody diluted 1:1000 in LI-COR TBS intercept antibody diluent with Tween20 (cat. no. 927-65001) then blots were washed 3X with TBST and incubated with a LI-COR donkey anti-rabbit secondary conjugated to IRdye 800CW at 1:15,000 (cat. no. 925-32213, lot no. D20119-01) for 45 min then washed 4X with TBST. Blots were imaged on an Odyssey CLx.

To confirm that HPV18 E6 and E6AP antibodies recognized the correct proteins in HeLa cells we performed an siRNA knockdown experiment. HeLa cells were plated and after 24 h treated with siRNA against HPV18 E6 (Santa Cruz, cat. no. SC-270681, lot no. H1222), E6AP (Santa Cruz, cat. no. SC-43742, lot no. H1213) or a control siRNA (Santa Cruz, cat. no. SC-#37007, lot no. I1821) according to the Lipofectamine RNAiMAX (cat. no. 13778075) protocol. Cells were allowed to grow for 48 h, and lysates were prepared as described above. 30 μg of lysates were used for the SDS-PAGE and the blot was probed with HPV18 E6 antibody (Genetex, cat. no. GTX132687, lot no. 42592) and anti-E6AP (Sigma, cat. no. E8655, lot no. 0000137246), each at a 1:1000 dilution, overnight at 4 °C. Blot was washed 4X 10 min. with TBS-T, incubated with LI-COR donkey anti-rabbit secondary conjugated to IRdye 800CW (LI-COR, cat. no. 925-32213, lot no. D20119-01) and LI-COR donkey anti-mouse conjugated to IRdye680RD (LI-COR, cat. no. 925-68072, lot no. D20125-11) at 1:10,000 for 1 h, washed 4X 10 min. with TBS-T and imaged on an Odyssey CLx. After which blots were stripped with NewBlot Nitro Stripping Buffer (LI-COR, cat. no. 928-40030), and re-probed with anti-GAPDH (Thermo, cat. no. MA5-15738, lot no. WL332983) antibody diluted 1:1000 overnight and incubated at 4 °C. Blot was washed as above and then probed with donkey anti mouse IRdye800CW (LI-COR cat. no. 925-32212, lot no. D20412-01) at 1:10,000 dilution for 1 h at RT, then washed and imaged as above.

## Fluorescent protein standards
We generated fluorescent protein standards for use in sedimentation gradients as follows: equine skeletal muscle myoglobin (Sigma, cat. no. M0630-250mg, source #SLCL1982; 6.3 mg), gamma globulin from bovine blood (Sigma, cat. no. G7516-1G, lot no. SLBZ8713; 7.3 mg), phosphorylase B from rabbit muscle (Sigma, cat. no. P6635-5mg, lot #SLCF9637; 2.3 mg), bovine serum albumin (Sigma, cat. no. A9418-5G, source #SLCM6875; 10 mg), and chicken egg white albumin (ovalbumin; Sigma, cat. no. A7641-250mg, source no. SLBL9222V; 6.8 mg) were individually resuspended in 400 μL 1X PBS. Atto-488 NHS ester (Sigma, cat. no. 41698-1MG-F; 1 mg) was resuspended in 500 μL of DMSO, and 100 μL of the dye was added to each protein slurry. The reaction was allowed to run for 1 h at room temperature with mixing. The protein was desalted using 2 mL Zeba spin desalting columns with 7 K MWCO (Thermo, cat. no. 89890), and the desalted proteins were diluted and combined in 1X PBS such that each protein was at a concentration of 1 mg/mL. We assumed 100% recovery from the desalting column. Atto-488 Fluorescent protein standard was visualized by SDS page using the 4–20% criterion gel followed by fluorescent imaging on an Amersham ImageQuant 800 using the Cy2 settings. A commercially available protein ladder (LI-COR, cat. no. 928-60000) was also scanned with IRshort and IRlong settings.

## Sucrose sedimentation gradients
Sucrose gradients were formed using 20% and 60% weight/volume solutions of sucrose dissolved in 25 mM Tris pH 7.4, 150 mM NaCl and 5% glycerol supplemented with 1X Halt Protease Phosphatase inhibitor cocktail (Thermo, cat. no. 1861284). Gradients were formed in open cap 5 mL tubes (Seton Scientific, cat. no. 7022) using the SW50 short

cap 20–60% sucrose W/V program on a biocomp gradient station model 153.

Lysates for centrifugation were prepared by lysing 70–80% confluent cells with cold Pierce IP lysis buffer (cat. no. 87787) on ice for 20 minutes. Plates were then scraped, and material was spun at max speed in a bench top centrifuge for 20 min. at 4 °C. Supernatant was removed and lysates quantified using a Pierce 660 assay (cat. no. 1861426). 500–1000 μg of lysate along with 20 μL of Atto-488 labeled protein standard was pipetted onto the surface of the sucrose column. Gradients were spun in a SW55 Ti rotor at 367,600 $g$ for 24 h at 4 °C. 200 μL fractions were collected using a biocomp piston gradient fractionator model 152.

Fractions were concentrated via methanol chloroform precipitation. In short, fractions were mixed with 800 μL of methanol, vortexed, 200 μL of chloroform was added followed by vortexing. 600 μL of $H_2O$ was added and fractions were vortexed again. Fractions were spun at 14,000 $g$ for 2 min. Aqueous layer was removed and 800 μL methanol added followed by vortexing. Fractions were then spun at 14,000 $g$ for 3 min. after which methanol was removed and the precipitated protein dried in a speedvac. Fractions were resuspended in 1X LI-COR protein loading buffer (cat. no. 928-4004).

One fifth of each fraction was resolved on a 4–20% TGX criterion gel and Atto488 labeled molecular weight standards were visualized on an Amersham ImageQuant 800 as described above. Blots were then transferred to nitrocellulose, blocked with TBS intercept blocking buffer (LI-COR, cat. no. 927-65001) then incubated overnight with anti HPV18 E6 (Genetex, cat. no. 132687) antibody diluted 1:1000 in LI-COR TBS intercept antibody diluent with Tween20 (cat. no. 927-65001) at 4 °C. Blots were washed 3X with TBST and incubated with a LI-COR donkey anti-rabbit secondary conjugated to IRdye 800CW at 1:15,000 (cat. no. 925-32213) for 45 min. then washed 4X with TBST. Blots were imaged on an Odyssey CLx. Blots were then incubated overnight at 4 °C with anti-E6AP (Sigma, cat. no. E8655) antibody diluted to 1:1000 in LI-COR intercept antibody diluent, blots were washed 3X with TBST and incubated with a LI-COR donkey anti mouse secondary conjugated to IRdye 680RD at 1:15,000 (cat. no. 925-68072) for 45 min. then washed 4X with TBST. Blots were again imaged on an Odyssey CLx. Image Studio version 5.2 was used to quantify band intensity of HPV18 E6 and E6AP. The total signal percentage in each lane was determined by summing up the protein signal detected in all lanes. The fluorescent standard band intensity was quantified using the ImageJ gel analysis plug-in. To normalize the fluorescent standards, the signal in each fraction was divided by the maximum fraction signal detected in the fraction for that protein. To align fractions across different experiments, the peak myoglobin fraction from each gradient was set as the fifth fraction.

## Semi-quantitative analysis of HPV18E6 and E6AP levels in lysate
Cells were grown to 70–80% confluency before being harvested and lysed with cold Pierce IP lysis buffer (cat. no. 87787, lot # WK337252) supplemented with 1X Halt Protease Phosphatase inhibitor cocktail (Thermo, cat. no. 1861284, lot # XC342080) by incubating on ice for 10 min. with intermittent mixing. Lysates were cleared by centrifugation at maximum speed in a bench top centrifuge for 10 min. at 4 °C and subsequently quantified using a Pierce 660 assay (Thermo, cat no 22660, lot no. XA338566). 5–60 μg of each HeLa and HT1080 lysate and 0.47–15 ng of purified, recombinant MBP-18E6 and E6AP were resolved on a 4–20% TGX Criterion gel before being transferred to nitrocellulose. Membranes were subsequently blocked with TBS Intercept blocking buffer (LI-COR, cat. no. 927-65001) and incubated overnight with anti-HPV18 E6 (Genetex, cat. no. 132687) or anti-E6AP (Cell Signaling, cat. no. D10D3, lot no. 7526S) diluted 1:1,000 in LI-COR TBS intercept antibody diluent with Tween20 (cat. no. 927-65001) at 4 °C. Blots were washed with TBS-T and incubated with donkey-α-rabbit (800 nm conjugated; LI-COR, cat. no. 925-32213, lot no. D20119-01) secondary antibody for 2 h at RT with shaking, then washed

with TBS-T four times with shaking. Blots were imaged with an ODYSSEY CLx LI-COR instrument using Image Studio software (version no 5.2) and gel bands were quantified using auto-exposure, and background subtraction was carried out using the software. The fluorescence intensity for the recombinant protein bands was plotted against protein amount (converted to femtomoles) and a standard curve was generated. The fluorescence intensity for either HPV18 E6 or E6AP present in lysate was also quantified, and the amount of protein corresponding to the fluorescence intensity was interpolated from the standard curve utilizing GraphPad PRISM software. The amount of interpolated protein from the standard curve was normalized to the amount of lysate input to obtain an approximate amount of protein in femtomoles per microgram of lysate.

### Monitoring the stability of the MBP-16E6(4C4S) and E6AP complex with peptide 13

To investigate the stability of the MBP-16E6(4C4S) and E6AP complex in vitro a covalent peptide, peptide-13 (Ye et al. Chemical Science 2023), was used. MBP-16E6(4C4S) at a concentration of 2 µM was either incubated alone or co-incubated with E6AP (2 µM) in a buffer solution composed of 1X PBS, 5% glycerol, and 1 mM DTT. This mixture was allowed to rest for 15 min. on ice to ensure effective complex formation. Subsequently, peptide-13 was introduced to the solution at a final concentration of 10 µM. The samples were then incubated at 4 °C for up to 72 h. Samples were extracted from this mixture at 24 h intervals. Following extraction, samples were subjected to intact mass spectrometry to determine the level of peptide-13 conjugation to MBP-16E6.

### Binding constant determination of MBP-16E6 and E6AP by mass photometry measurement

E6AP (100 nM) and MBP-16E6(4C4S) (25, 50, or 100 nM each) were diluted from stock and resuspended in 1X PBS supplemented with 1 mM DTT. Mixtures were incubated for 1 h at room temperature. Measurements were performed on the Two$^{MP}$ mass photometer (Refeyn Ltd) as previously[80]. Briefly, a silicone sample well cassette and glass carrier slides (Refeyn Ltd) were assembled and placed on the sample stage. In an unused well, 18 µL of buffer was added to find focus, after which 2 µL of E6AP, MBP-16E6(4C4S), or mixture was mixed in homogeneously. The final concentration of the measurement was 10-fold diluted from the mixture. Data was recorded for 1 min. To determine the binding affinity, data was analyzed using DiscoverMP v2023 R2 (Refeyn Ltd). Intervals were set based on the mass of E6AP, MBP-16E6(4C4S), and E6AP:MBP-16E6(4C4S) complex to quantify the particles belonging to each species in the 1 min. movie. Particle counts were converted to concentrations of each species at equilibrium as previously described[81]. In summary, particle counts were converted to concentrations of each species at equilibrium using the following equations described in[81] where the $K_D$ was determined by the following equilibrium equation (1).

$$A + B \rightleftharpoons AB$$

$$A_{conversion} = \frac{[A]_{initial}}{\sum (A_{counts} + AB_{counts})}$$

$$B_{conversion} = \frac{[B]_{initial}}{\sum (B_{counts} + AB_{counts})}$$

$$[A]_{eq} = A_{conversion} * A_{counts}$$

$$[B]_{eq} = B_{conversion} * B_{counts}$$

$$[AB]_{eq} = [B]_{initial} - [B]_{eq}$$

$$K_D = \frac{[A]_{eq}[B]_{eq}}{[AB]_{eq}}$$

The $K_D$ was calculated from the concentrations of each species at the equilibrium state. Results from three different conditions at triplicates were averaged and the standard deviations were reported[82].

### Reporting summary

Further information on research design is available in the Nature Portfolio Reporting Summary linked to this article.

## Data availability

A Source Data File is provided with this paper. The structure file data generated in this study have been deposited in the PDB database under accession code 8GCR and at the EMDB under EMD-29941. All data are available from the corresponding author. The MD simulation files, including the initial system configurations, final states, trajectories, simulation parameters, and force field parameters, have all been deposited. For the five systems studied in this paper (Supplementary Table 2), the files can be retrieved at Zenodo (https://zenodo.org/) using the following link [https://doi.org/10.5281/zenodo.10622192] as "Wang_et_al_Nat_Commun_2024_E6AP_MD.zip". Source data are provided with this paper.

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

## Acknowledgements

Houxia Shi (Biortus Co Ltd, Wuxi, China) assembled the ternary protein complex used for the cryoEM study. Shijun Zhang and Jack Yan (Biortus Co Ltd, Wuxi, China) collected the electron density maps, solved the initial models, and reconstituted the ternary complex structure. We would like to thank Janani Sridar for project support. We thank Ritu Shrestha of Refeyn for designing and executing the mass photometry experiments and Amy Chau of Refeyn for data analysis and discussions (info@refeyn.com). Additionally, we would like to thank Manoj Rathinaswamy and Breane Budaitis (Calico Life Sciences LLC) for their critical evaluation of the manuscript. Calico Life Sciences LLC provided funding for all resources and salaries for this work.

## Author contributions

Ub assays were conducted by J.C.K.W., Z.J, and H.B., while sedimentation gradients were run by I.F. and H.B. M.B. deposited the PDB files, and J.C.K.W. produced HPV16 E6 proteins and conducted SPR assays. T.-Y. Lin conducted NanoBRET assays under supervision from M.P.L. A.M. conducted molecular dynamics simulations. A.H.N., Q.H., D.S., and D.E. oversaw the project. A.H.N. and Q.H. conceived of the project. A.H.N., Q.H., H.B., and A.M. wrote the manuscript and made the Figures with input from all authors.

## Competing interests

All authors are employees of Calico Life Sciences LLC.
