## [Peer Review File · Nature Communications]

Structure of the HPV16 p53 degradation complexREVIEWER COMMENTS

Reviewer #1 (Remarks to the Author):

In their work, Wang et al. describe the 3D structure of a complex consisting of the HPV16 E6 oncoprotein, full-length E6AP and the core domain of p53. The structure (3.3 Å resolution) was solved by cryo-EM and is complemented by MD simulations to further investigate E6-E6AP interface contacts and the conformations of free and bound E6AP. These results bring crucial novel insight by revealing an additional E6-E6AP interface, which was not observed in the previous structures. This additional interface is dissected by mutagenesis experiments coupled to binding and ubiquitination studies.

Overall, the authors have performed a rich and accurate work. The results disclosed represent an important step forward towards the understanding of the mechanisms leading to E6-mediated degradation of p53. However, I have several major issues that are listed below.

1. The first and most important point concerns the K_D value of the HPV16 E6/full-length interaction, which is found to be 144 pM, which is discussed several times throughout the manuscript. There is no doubt that the E6 binding affinity for full-length E6AP is higher compared to that for the LxxLL peptide. Yet, I am quite skeptical about the claim of a K_D value of 144 pM. If this was true, E6AP would capture most E6 in HPV-positive cancer cells, which is definitely not the case since the last two decades of research in the HPV field have shown that high-risk mucosal E6 proteins have numerous E6AP and p53 independent functions. I wonder whether the SPR approach used based on the use of S CAP sensor chips leads to over-estimation of the affinity constants. The S CAP sensor chip uses hybridized oligos and E6 has high background affinity for oligonucleotides. Therefore, the authors should determine the binding affinity of the HPV16 E6/full-length interaction by an alternative biophysical technique, such as ITC or fluorescence polarization transfer (competition studies using a fluorescently labelled peptide). If the authors find a big discrepancy with the SPR values, they should repeat measurements on the interactions involving mutant E6 and E6AP proteins.
2. If SPR data will be kept in the paper, the authors should carefully describe the experimental conditions. What are the control conditions? And the concentrations of injected MBP-E6? Are they always the same for all experiments (usually they are adjusted according to the binding affinities)? Please also show raw data for the control conditions.
3. The authors say that recombinant E6AP is a monomer according to SEC analysis. Is there any evidence for E6AP oligomers in solution (eg. from light scattering measurements) that might break off on the gel filtration column?
4. Supplementary Figures 3 and 12: the SEC profiles should be plotted with volume on the x-axis and not time.
5. Could the authors comment a bit more on the particle/conformational homogeneity/heterogeneity in the cryo-EM analyses? I wonder whether the MBP tag might partially hinder the formation of the

additional E6-E6AP interface, since in the previous E6/E6AP LxxLL peptide/p53 structure, MBP tag contacts a similar surface of E6.

Minor issues:

-Page 5 third paragraph: please refer to the “C-terminal lobe of the HECT domain” and not to the “C-terminal HECT domain”.

Reviewer #2 (Remarks to the Author):

Review: Structure of the HPV16-p53 Destruction-Complex,

JCK Wang, HT,Baddock, IT Foe, A Mafi, M Bratkowski, D Stokoe, D Eaton, Q Hao, AH Nile.

Accept with revisions.

The human papilloma virus induces oncogenic cellular transformations in infected mammalian cells through the 16E6 protein directed destruction of p53 by recruiting p53 to the ubiquitin ligase E6AP (UBE3A). This manuscript presents the structure of the fullest complex examined to date of this p53 destruction-complex generated by combining the full-length HPV 16E6 protein, E6AP ubiquitin ligase, and p53core (S94-T312). The work by JCK Wang et al. identifies three key regions of interaction between 16E6 and E6AP from this structural investigation and defines their contributions to the pico-Molar affinity between 16E6 and E6AP through site directed mutagenesis as measured by surface plasmon resonance. These altered interactions have a requisite functionally reduced level in p53 and E6AP ubiquitination due to the reduced affinity between 16E6 and E6AP. These findings are presented as an important step toward directed therapeutic development for HPV derived cancer intervention, but no small molecule inhibitors are employed in this study and it is acknowledged vaccines have hampered therapeutic development due to profitability concerns by pharmaceutical companies. The extensive analysis of the p53, 16E6, E6AP destruction complex presented by this work is significant and informative.

Major points

(1) Although an atomic model was generated from a very clean 3.38 Angstrom coulombic map, it is not appropriate to state this is an atomic resolution map as the density does not resolve individual atoms and is far from 2.0 Angstrom, the resolution which is typically taken to be atomic resolution (PMID: 33087927). All statements of this nature should be removed as it is misleading. Correcting these statements will place in context the true nature of the structural fitting of atomic coordinates. This reviewer does not think fitting at this resolution with previous determined structures, MD, and alpha

fold as guidance is untoward, but stating the work represents atomic resolution is not correct. Indeed, mutational analysis and physiological ramifications are strong support for the fitted models and makes for a solid study.

(2) The HECT domain was stated as poorly resolved in the cryoEM map. The map projections in the PDB validation report suggest density is present for the HECT domain albeit weaker. Why was no attempt at 3D variability analysis reported? Since 3D conformational variability tools are readily available in today's single particle reconstruction software (PMID: 33582281, PMID: 29856314), it would be expected that this catalytic domain of the E6AP protein would be an informative part of this structural study. An explanation as to why this was not pursued or reported would be informative.

(3) Molecular dynamic simulations predict the LXXLL helical motif of E6AP is unstructured in the apo-state, and only resolves as a helix when bound to the 16E6 protein as seen in the cryoEM map and other studies using a short LXXLL peptide bound to 16E6. However, stating these simulations "confirms" the hypothesis of the two states existing without direct observation or further biochemical experimentation (page 8, line 31) is too strong. Substituting "supports" for "confirms" would be more fitting.

(4) Figure 2 is titled "Protein-protein interactions between 16E6, E6AP, and p53core, but no detailed information is presented in the figure regarding p53 core interaction with either 16E6 and E6AP, even though there does appear that within site 1 some p53core interactions present; this is actually presented in Figure 4a and Supplementary Figure 16a. A more appropriate title may be "Three primary sites drive 16E6 and E6AP interaction" and would more contextually match the text within the manuscript's results section that refers to figure 2.

(5) The discussion relating to therapeutic interventions on blocking 16E6 and E6AP are brief at best and could benefit from deeper exploration particularly regarding how to interrupt a 143.5 pM interaction and how this study would define a path forward.

Minor points

Page 8, line 10: CryoEM is duplicated

Page 9, Line 16: the line needs right justification

Figure 5: depected should be depicted.

Reviewer #3 (Remarks to the Author):

In this manuscript, Wang et al reported a combined single-particle cryo-EM and all-atom MD simulation study on the full-length E6AP protein in complex with HPV16 E6 and p53. They presented the crystal structure of E6AP-E6-p53 complex, analyzed the extensive interactions between E6AP and E6, and revealed the recognition mechanism of p53 by E6 and E6AP complex. The results provide some new structural insights into the therapeutic strategies for the treatment of HPV induced cancers.

More comments:

1. Page 5 and Fig S5: To better demonstrate the similarity between their solved crystal structure of E6AP-E6-p53 complex and the previously-reported E6AP-E6-p53 complex (Nature 2016), the authors are suggested to provide the RMSD values between those two structures as well as the RMSD values for the isolated E6AP LXXLL motif, E6 and p53.

2. Page 5 and Fig S8a: It was stated in the manuscript that the HECT domains were not resolved because of their significant flexibility. This is not convincing, because 1) the RMSF value of the HECT domains in Fig. S8 is quite similar to the structure-resolved regions (residues 250-750) which have low flexibility; 2) the confidence of the HECT domains in AF2-predicted structure is rather high ($70 < pLLDT < 90$), indicating that the HECT domains are structurally conserved and have stable structure. This issue needs to be clarified.

3. Two AF2-predicted structures were presented in this manuscript (Fig S8a: AF-Q05086-F1-model_v4 and Fig S9a: AF-Q05086-F1-model_v2.pdb). The first predicted structure was selected for MD simulation of apo-E6AP, and the second predicted structure was selected as a comparison with the experimental E6AP structure. What are the reasons for choosing different AF2 predicted structures? What will the results be if the two different AF2-predicted structures are both used for MD simulations and structural comparison? In addition, why the other AF2-predicted structures were not considered in the manuscript, such as AF-Q05086-F1-model_v1 and AF-Q05086-F1-model_v3?

4. Page 7 and Fig S10b-d: To better characterize the stability of three sites of interactions between E6AP and E6, the authors are suggested to add a new panel in Fig S10 by providing the time evolution of the fraction/percentage of the E6AP-E6 native contact at each interaction site.

5. Page 7 and Fig S10a: The authors calculated the time evolution of the RMSD values of each monomer in E6AP-E6-p53. This reviewer suggests the authors also calculate the RMSD values of the entire E6AP-E6-p53 complex, which can provide the overall structural stability of E6AP-E6-p53 complex.

6. There are several recent publications which are closely related to the content of this manuscript. Here are some examples, [Nat Rev Drug Discov 22, 127-144 (2023)], [Biophys J 121, 1704-1714 (2022)], [Nat Commun 12, 986 (2021)], [Pharmacol Res 177, 106128 (2022)], [Sig Transduct Target Ther 8, 92 (2023)]. These papers need to be discussed/cited in this manuscript.

7. Fig S8: There is a spelling error in “simulaiton”, which should be corrected.

October 23rd, 2023

Dear Reviewers,

Outlined below are the key changes we incorporated:

Major changes:

- Three additional authors were added to the manuscript to assist in the requested experiments (Molly Lin, Zena D. Jensvold, and Magdalena Preciado López)
- We applied mass photometry as an orthogonal method to monitor the MBP-16E6 and E6AP binding affinity (**Figure S5**).
- We identified E6AP residues R417 and R418 through MD simulations which can be mutated to decouple p53 ubiquitination without compromising E6AP autoubiquitination (**Figure 6**).
- We developed a novel recombinant NanoBRET assay as an orthogonal measurement of MBP-16E6 and E6AP binding affinity (**Figures S6, S19**).
- We applied our recently reported 16E6-directed “reactide” to probe the MBP-16E6, and E6AP complex stability (**Figure S16**).
- We demonstrated that 18E6 in HeLa cells is found in molar excess to E6AP (**Figure S30**).
- Throughout, figure numbering was significantly altered.
- Main Figures:
 - **Figure 3**: transitioned data from supplemental materials to a main figure.
 - **Figure 4**: relocated data from supplemental materials to a main figure.
 - **Figure 6**: new data highlighting the impact of newly identified/tested double E6AP mutants (see above).
- Supplementary Figures:
 - Figures including S4, S5, S6, S7, S16, S18 through S33 contain new data, transformations from the main figures, and various requested amendments, as detailed in the reviewer commentary.

Textual Changes: We significantly revised sections of the text to offer more clarity and address specific concerns voiced by the reviewers. Significant changes are demarcated in purple for ease of reference.

Minor Textual Edits: Apart from the major changes, we also fine-tuned our manuscript: rectified typographical errors and improved visuals across figures for better alignment, uniform font application, and legend clarity.

Thank you for taking the time to review our manuscript.

Sincerely,

Aaron H. Nile (aaronnile@calicolabs.com)

Principal Investigator, Oncology

Qi Hao, Ph.D. (qhao@calicolabs.com)

Principal Scientist 2, Associate Director of Protein Sciences

REVIEWER COMMENTS

Reviewer #1 (Remarks to the Author):

In their work, Wang et al. describe the 3D structure of a complex consisting of the HPV16 E6 oncoprotein, full-length E6AP and the core domain of p53. The structure (3.3 Å resolution) was solved by cryo-EM and is complemented by MD simulations to further investigate E6-E6AP interface contacts and the conformations of free and bound E6AP. These results bring crucial novel insight by revealing an additional E6-E6AP interface, which was not observed in the previous structures. This additional interface is dissected by mutagenesis experiments coupled to binding and ubiquitination studies.

Overall, the authors have performed a rich and accurate work. The results disclosed represent an important step forward towards the understanding of the mechanisms leading to E6-mediated degradation of p53. However, I have several major issues that are listed below.

We thank the reviewer for the encouraging remarks and are pleased that the reviewer finds this work to be “rich and accurate work” and represents an “important step”. We've carefully addressed each of the reviewer's comments in the subsequent sections below.

1. The first and most important point concerns the KD value of the HPV16 E6/full-length interaction, which is found to be 144 pM, which is discussed several times throughout the manuscript. There is no doubt that the E6 binding affinity for full-length E6AP is higher compared to that for the LxxLL peptide. Yet, I am quite skeptical about the claim of a KD value of 144 pM. If this was true, E6AP would capture most E6 in HPV-positive cancer cells, which is definitely not the case since the last two decades of research in the HPV field have shown that high-risk mucosal E6 proteins have numerous E6AP and p53 independent functions. I wonder whether the SPR approach used based on the use of S CAP sensor chips leads to over-estimation of the affinity constants. The S CAP sensor chip uses hybridized oligos and E6 has high background affinity for oligonucleotides. Therefore, the authors should determine the binding affinity of the HPV16 E6/full-length interaction by an alternative biophysical technique, such as ITC or fluorescence polarization transfer (competition studies using a fluorescently labeled peptide). If the authors find a big discrepancy with the SPR values, they should repeat measurements on the interactions involving mutant E6 and E6AP proteins.

We were also surprised by the high affinity interaction between MBP-16E6(4C4S) and full-length E6AP, which was the motivation behind solving the MBP-16E6, full-length E6AP, and p53 core ternary complex. Although the data is not included within this manuscript, our collaborators have also confirmed this high affinity interaction by SPR.

In response to Reviewer 1's statement: "If this was true, E6AP would capture most E6 in HPV-positive cancer cells," we appreciate the body of research from the past two decades indicating that E6 proteins possess multiple E6AP-independent activities. However, a crucial question arose based on the Reviewer's question: **What is the stoichiometry between HPV E6 and E6AP under steady-state conditions?** As far as we know, their relative stoichiometry is unreported.

To address this, we utilized HPV18+ HeLa cells, choosing them over HPV16+ cells due to the poor specificity of commercially available 16E6-directed antibodies (e.g., Thermo: MA146057, PA5-117355, BS-0990R, BS-2968R, Genetex: GTX132686, Santa Cruz: SC-460, Abbiotec: 251401). For a control, we employed HT1080 (HPV-) cells to assess the relative E6AP expression levels in a generic cancer cell line. We used a semi-quantitative western blot method, applying recombinant MBP-18E6 or full-length E6AP proteins as internal standards to quantify the absolute amounts of the endogenous proteins. Our findings indicate that 18E6 is present in a molar ratio at ~2.9-fold higher than E6AP (**Supplemental Figure 30**). The 18E6 molar excess implies that "free" 18E6 remains, which could be accessible for interactions with previously reported binding partners. Ideally, we would introduce a tag (e.g., GFP, FLAG) at the 18E6/16E6 endogenous locus for direct comparison; however, this is impractical given the reported ~12-copies of 18E6 in HeLa cells (PMID: 23925245).

In response to Reviewer 1's second concern regarding the potential over-estimation of affinity constants using the S CAP sensor chip due to E6's known affinity for oligonucleotides: to mitigate potential bias introduced by the S CAP oligo chips, we quantified the K_D between MBP-16E6(4C4S) and biotin-E6AP using the standard SA chips, which do not contain DNA oligos. We previously tested these chips but employed the S CAP chips to reduce experimental costs (i.e. one chip per experiment given the high affinity interaction). We employed single cycle kinetics with the standard SA chips since we could not find a regenerative protocol with a quasi-permanent E6AP immobilized surface that did not compromise the integrity of the immobilized E6AP in-between analyte injections. Consistent with the results from the S CAP chips, the affinity measured with the standard SA chips was 200 pM (**Figure S4b**), which is in close agreement with the 140 pM observed using the S CAP chips (**Figure 1a**). These results lend support to our initial findings, indicating that the high affinity interaction was not influenced by the DNA oligomer attachment strategy employed by S CAP SPR-chips.

In response to Reviewer 1's third suggestion, regarding the validation of the HPV16 E6/full-length interaction binding affinity through an alternative biophysical technique: to this end, we tested multiple orthogonal assays to verify the binding constant between MBP-16E6 and E6AP. This included ITC, nanoBRET, and mass photometry.

Regrettably, our attempts with ITC did not yield consistent measurements. It has been reported that apo-16E6 is flexible and undergoes structured conformational changes upon LXXLL-binding (and likely E6AP). Traditional ITC models may not accurately reflect these binding events. Consequently, we have refrained from discussing these in the main text; however, for transparency, we included a representative ITC trace below. Since we were not confident in the data quality (K_D is likely below 10 nM) we did not include this in the manuscript.

Continuing this effort, we built a NanoBRET system inspired by Promega's in-cell NanoBRET system composed of recombinant MBP-16E6-Halo protein as a BRET acceptor and E6AP-Nluc as a BRET donor. Using this system, we found the K_D to be approximately 15.27 nM (see **Figures S6** and **S7**). This represented a 106-fold reduction in K_D which we thought was likely a result of large protein tags required for this system (19 and 33 kDa, respectively; see **Figure S6a**). Evaluation of these proteins by SPR demonstrated a ~20-fold reduction in binding affinity ($K_D = 4$ nM) from the parental proteins (**Figure 1a** vs. **Figure S7**).

After normalizing for this reduced affinity in the MBP-E6-Halo and E6AP-Nluc pairing, the discrepancy in K_D between the NanoBRET and SPR measurements reduced to ~5-fold. This comparative analysis further supports that the SPR measurements faithfully capture the true affinity between 16E6 and E6AP. To better understand this system, we subjected known MBP-16E6 mutants that interfere with E6AP binding to NanoBRET. The results were consistent with our original observations using SPR and p53 ubiquitination assays (**Figures S19**, **S32**, and **S33** vs. **Figures 5**).

As an orthogonal method we used mass photometry, a label-free light scattering technique to monitor complex formation (Young, Hundt et al. 2018) with the help from Refeyn as we do not have direct access to the equipment. We incubated varying concentrations of MBP-16E6(4C4S) (2.5 nM, 5 nM, or 10 nM) and E6AP (10 nM). This

provided a clear readout of dimeric complex formation (**Figure S5a-b**). This system is not optimized for pM concentrations of proteins due to sensitivity limitations; however, the data supports a binding affinity between MBP-16E6 and E6AP between 99 pM to 1.29 nM (**Figure S5**) further supporting the accuracy of SPR measurements.

We recently reported the characterization of a covalent “reactide”, Pep13, aimed at targeting Cys58 within the LXXLL hydrophobic cavity of 16E6 (i.e., the “site 1” interface of the 16E6:E6AP interaction). This was initially published in bioRxiv (Ye, Zhang et al. 2023) and subsequently accepted by Chemical Science (DOI: 10.1039/d3sc02782a, not yet published as of 2023-10-20). Pep13, has an apparent K_i of 17 nM and is based on the LXXLL motif from E6AP modified at its N- and C-termini with the incorporation of unnatural amino acids to improve affinity and Dha to react with Cys58 in the hydrophobic cavity of 16E6 (Cys58 is not conserved in HPV18). We were curious if Pep13 could recognize recombinant MBP-16E6 when in a complex with E6AP (essentially, could it disrupt the dimeric complex?). After a three-day incubation of the MBP-16E6:E6AP complex at 4°C did not yield a pep13 adduct on MBP-16E6 when 16E6 and E6AP were complexed. However, an adduct was observed when pep13 was incubated with MBP-16E6, with near complete conjugation occurring by 24 h. While this observation doesn't yield a binding affinity, it underscores that the LXXLL binding site within the dimeric 16E6:E6AP complex remains shielded, even after an extended time (**Figure S16**) consistent with our molecular dynamics simulations (**Figure 3b**) and biophysical measurements described in this manuscript.

In summary to Reviewer 1's first comment, our findings are summarized below:

1. The K_D of the 16E6 and E6AP complex is between 140-200 pM, independent of the SA or S CAP SPR chip used (**Figure S4**).
2. In HeLa cells, 18E6 is in molar excess to E6AP which would allow for non-E6AP binding partners (**Figure S27**).
3. We were unable to generate technically convincing ITC data and will not include it in the final submission (see description above).
4. When corrected for the intrinsic reduction in binding affinity between 16E6-Halo and E6AP-nanoLuc used in the NanoBRET assay, we observed a ~5-6 fold-lower than reported by SPR. This data provides additional evidence that SPR faithfully captures the binding affinity between 16E6 and E6AP.
 - a. To further validate this system, we introduced multiple point mutations into 16E6 that disrupt the NanoBRET signal (**Figure S19, S32, S33**).

5. Mass photometry revealed a 1:1 binding interaction between MBP-16E6 and E6AP ranging from 97 pM to 1.29 nM (**Figure S5**).
6. A high-affinity covalent peptide (with $K_i = 17$ nM) was unable to recognize a pre-formed 16E6:E6AP dimeric complex—even after three days (**Figure S16**).
7. We move our molecular dynamics simulations to main figures which further highlights and supports a stable interaction between E6AP and 16E6 (**Figures 3 and 4**).

We hope that these supplementary experiments address Reviewer 1's concerns regarding the high affinity interaction between MBP-16E6 and E6AP and provide sufficient data that SPR faithfully captures the high affinity interaction between 16E6 and E6AP.

2. If SPR data will be kept in the paper, the authors should carefully describe the experimental conditions. What are the control conditions? And the concentrations of injected MBP-E6? Are they always the same for all experiments (usually they are adjusted according to the binding affinities)? Please also show raw data for the control conditions.

We have expanded the experimental description of our SPR conditions within the **Materials and Methods** section. We've described reference cell subtraction data against a flow cell that does not have immobilized E6AP on its surface. Additionally, we added the data of wild type 16E6(4C4S) injection onto immobilized surface each day an experiment was carried for SPR analysis to ensure that the E6AP still retained full binding functionality and that any decrease in affinity we measured using our 16E6 mutants was not due to the quality of E6AP now shown in Supplemental **Figure S31**.

To address the reviewer's comment on the concentrations chosen for injected MBP-16E6, we included **Figure S4a** to show that concentrations of MBP-16E6 above 50 nM causes significant binding to the dextran surface of the Biacore sensor chips. While we would ideally like to go higher in concentration for the lower affinity mutants, the background dextran binding limited our concentration range.

3. The authors say that recombinant E6AP is a monomer according to SEC analysis. Is there any evidence for E6AP oligomers in solution (eg. from light scattering measurements) that might break off on the gel filtration column?

We did not see evidence of E6AP oligomers when the protein is recombinantly expressed by Sf21 insect cells. The E6AP was well-behaved and showed good homogeneity during purification. A final SEC-MALS analysis of our purified E6AP aligns with a monomeric molecular weight. To address the reviewer's curiosity we have

included the MALS measurement below.

4. Supplementary Figures 3 and 12: the SEC profiles should be plotted with volume on the x-axis and not time.

We have plotted the values using volume and not time (see Figures S18, S24, S28, and S32).

5. Could the authors comment a bit more on the particle/conformational homogeneity/heterogeneity in the cryo-EM analyses? I wonder whether the MBP tag might partially hinder the formation of the additional E6-E6AP interface, since in the previous E6/E6AP LxxLL peptide/p53 structure, MBP tag contacts a similar surface of E6.

We did not observe density that could be attributed to MBP during particle analysis, suggesting that it does not impact complex formation. 16E6-MBP interfaces were observed in the previous 16E6-MBP-E6AP LxxLL/p53 crystal structure (PDB 4xr8, (Martinez-Zapien, Ruiz et al. 2016)). One crystal structure interface occurs between 16E6 near residue R84 (using the E6 numbering from this manuscript) and MBP residue T367 (please see the figure below). Note, in our cryo-EM structure, this 16E6-MBP interaction is not possible because a portion of E6AP near residue A549 is found in the same location as MBP (around residue T367).

A second interface from the crystal structure occurs around 16E6 residue K129 and MBP residue E275 (see the below image). This interface is also not possible in our cryo-EM structure as E6AP residue E472 would clash with MBP residue E279. Furthermore, part of MBP near residue A83 clashes with part of E6AP around T465. There are also several other MBP-E6AP clashes in both sites that are not shown. Therefore, the 16E6-MBP interfaces observed in the crystal structure appear incompatible with the full length 16E6/E6AP complex.

Overall, the previously observed 16E6-MBP interfaces are likely due to crystal packing. The MBP-E6AP LxxLL protein used in the crystal structure featured the C-terminus of MBP fused to the LxxLL peptide via a 3 alanine linker (Martinez-Zapien, Ruiz et al. 2016), artificially putting MBP in proximity to E6 upon LxxL binding. The construct used in this report features MBP fused to the N-terminus of 16E6 via a longer linker that is expected to be flexible (GGGGSENLYFQG; **Figure S3a**). We do not observe density for MBP or for the first 7 residues of 16E6, suggesting that MBP is very

likely flexible. Therefore, it is unlikely that MBP is contacting 16E6 in our structure.

Minor issues:

-Page 5 third paragraph: please refer to the “C-terminal lobe of the HECT domain” and not to the “C-terminal HECT domain”.

We have modified the phrasing in three locations within page 5 (Line 163-179).

Reviewer #2 (Remarks to the Author):

Review: Structure of the HPV16-p53 Destruction-Complex,

JCK Wang, HT, Baddock, IT Foe, A Mafi, M Bratkowski, D Stokoe, D Eaton, Q Hao, AH Nile.

Accept with revisions.

The human papilloma virus induces oncogenic cellular transformations in infected mammalian cells through the 16E6 protein directed destruction of p53 by recruiting p53 to the ubiquitin ligase E6AP (UBE3A). This manuscript presents the structure of the fullest complex examined to date of this p53 destruction-complex generated by combining the full-length HPV 16E6 protein, E6AP ubiquitin ligase, and p53core (S94-T312). The work by JCK Wang et al. identifies three key regions of interaction between 16E6 and E6AP from this structural investigation and defines their contributions to the pico-Molar affinity between 16E6 and E6AP through site directed mutagenesis as measured by surface plasmon resonance. These altered interactions have a requisite functionally reduced level in p53 and E6AP ubiquitination due to the reduced affinity between 16E6 and E6AP. These findings are presented as an important step toward directed therapeutic development for HPV derived cancer intervention, but no small molecule inhibitors are employed in this study and it is acknowledged vaccines have hampered therapeutic development due to profitability concerns by pharmaceutical companies. The extensive analysis of the p53, 16E6, E6AP destruction complex presented by this work is significant and informative.

We are pleased that the reviewer finds our work significant and informative, and we thank the reviewer for the encouraging comments.

Major points

(1) Although an atomic model was generated from a very clean 3.38 Angstrom coulombic map, it is not appropriate to state this is an atomic resolution map as the density does not resolve individual atoms and is far from 2.0 Angstrom, the resolution which is typically taken to be atomic resolution (PMID: 33087927; (Yip, Fischer et al. 2020)). All statements of this nature should be removed as it is misleading. Correcting these statements will place in context the true nature of the structural fitting of atomic coordinates. This reviewer does not think fitting at this resolution with previous determined structures, MD, and alpha fold as guidance is untoward, but stating the work represents atomic resolution is not correct. Indeed, mutational analysis and physiological ramifications are strong support for the fitted models and makes for a solid study.

We have removed the phrase "atomic resolution" in the revised manuscript to avoid confusion.

(2) The HECT domain was stated as poorly resolved in the cryoEM map. The map projections in the PDB validation report suggest density is present for the HECT domain albeit weaker. Why was no attempt at 3D variability analysis reported? Since 3D conformational variability tools are readily available in today's single particle reconstruction software (PMID: 33582281, (Punjani and Fleet 2021); PMID: 29856314, (Nakane, Kimanius et al. 2018)), it would be expected that this catalytic domain of the E6AP protein would be an informative part of this structural study. An explanation as to why this was not pursued or reported would be informative.

We thank the reviewer for allowing us to clarify this issue. Below we are depicting a map projection from the validation report to help illustrate our point:

Potential weak HECT domain density

While visible, the presumptive HECT domain from the above map projection is very weak in comparison to other areas. Modeling for this region was not possible due to the weak and ambiguous nature of the density (see red arrow). The cryo-EM team who worked on this structure indicated that 3D variability analysis would not be informative as the HECT domain density is too weak. Therefore, we have not included this analysis in our revision. However, to help clarify this point we included the above image (**Figure S13d**) to highlight this point and was referenced in the “Image Processing, 3D reconstruction” section of the **Materials and Methods** section (Line 621).

(3) Molecular dynamic simulations predict the LXXLL helical motif of E6AP is unstructured in the apo- state, and only resolves as a helix when bound to the 16E6 protein as seen in the cryoEM map and other studies using a short LXXLL peptide bound to 16E6. However, stating these simulations “confirms” the hypothesis of the two states existing without direct observation or further biochemical experimentation (page 8, line 31) is too strong. Substituting “supports” for “confirms” would be more fitting.

We have modified the text to downplay our confidence. It now reads, “Collectively, these results support the high flexibility of the isolated E6AP LXXLL motif is due to its lack of binding to 16E6” (Line 266). To keep with the spirit of this Reviewer’s comment we also modified the following sentence to “Notably, the E6AP LXXLL motif, specifically the E406-E415 in site 1, remains highly stable and maintains its alpha-helical structure with an RMSD value of $0.4 \pm 0.2 \text{ \AA}$, suggesting an important role of this LXXLL-motif mediated interface”. (Line 231)

(4) Figure 2 is titled “Protein-protein interactions between 16E6, E6AP, and p53core, but no detailed information is presented in the figure regarding p53 core interaction with

either 16E6 and E6AP, even though there does appear that within site 1 some p53core interactions present; this is actually presented in Figure 4a and Supplementary Figure 16a. A more appropriate title may be “Three primary sites drive 16E6 and E6AP interaction” and would more contextually match the text within the manuscript’s results section that refers to figure 2.

We adjusted the title of **Figure 2** to: “Three primary sites drive 16E6 and E6AP interaction”. Thank you for the suggestion – it is a better title.

(5) The discussion relating to therapeutic interventions on blocking 16E6 and E6AP are brief at best and could benefit from deeper exploration particularly regarding how to interrupt a 143.5 pM interaction and how this study would define a path forward.

We have expanded the discussion to discuss this in more depth colored in purple where changes were made.

Minor points:

Page 8, line 10: CryoEM is duplicated

Thank you for catching this. We removed the second CryoEM.

Page 9, Line 16: the line needs right justification

This is adjusted in the final text.

Figure 5: depected should be depicted.

Thank you for catching the error – it has been corrected.

Reviewer #3 (Remarks to the Author):

In this manuscript, Wang et al reported a combined single-particle cryo-EM and all-atom MD simulation study on the full-length E6AP protein in complex with HPV16 E6 and p53. They presented the crystal structure of E6AP-E6-p53 complex, analyzed the extensive interactions between E6AP and E6, and revealed the recognition mechanism of p53 by E6 and E6AP complex. The results provide some new structural insights into

the therapeutic strategies for the treatment of HPV induced cancers.

We thank the reviewer for review and suggestions. We have addressed each comment below and we hope that our responses are satisfactory.

More comments:

1. Page 5 and Fig S5: To better demonstrate the similarity between their solved crystal structure of E6AP-E6-p53 complex and the previously-reported E6AP-E6-p53 complex (Nature 2016), the authors are suggested to provide the RMSD values between those two structures as well as the RMSD values for the isolated E6AP LXXLL motif, E6 and p53.

Due to manuscript edits **Figure S5** is now **Figure S9**. As requested by the reviewer we included the direct comparison in **Figure S9a** and in the figure legend.

2. Page 5 and Fig S8a: It was stated in the manuscript that the HECT domains were not resolved because of their significant flexibility. This is not convincing, because:

1) the RMSF value of the HECT domains in Fig. S8 is quite similar to the structure-resolved regions (residues 250-750) which have low flexibility the confidence of the HECT domains in AF2-predicted structure is rather high ($70 < pLLDT < 90$), indicating that the HECT domains are structurally conserved and have stable structure. This issue needs to be clarified.

For greater clarity, we excluded the N-terminal region from our RMSF analysis of various E6AP domains. The resulting plot is presented in the figure below and in **Figure S12b**. This analysis reveals that the RMSF of the HECT domain shows significant fluctuations, reaching up to 10.0 Å, which indicates increased flexibility in this domain. It's worth noting that most regions exhibiting similar flexibility were not successfully resolved in the cryo-EM structure. Interestingly, the LXXLL motif peptide of E6AP also demonstrates a high degree of flexibility. But as captured by cryo-EM, binding to 16E6 stabilizes the LXXLL motif peptide into a very stable alpha helix with rmsd = 0.4-0.7 Å.

The HECT domain displays considerable flexibility while still retaining its structural integrity in terms of secondary structures, as shown in **Figure S12a,d**. Specifically, the HECT domain undergoes a body displacement of 11.5 Å relative to the E6AP core (stable Cα atoms of residues 126-170, 233-387, 395-432, and 439-758).

To understand this behavior, AlphaFold provides two key metrics: per-residue predicted local distance difference test (pLLDT) and the predicted aligned error (PAE). The pLLDT offers a confidence score that evaluates how reliably the protein folds, particularly in terms of its secondary structure. According to **Figure S12a**, the pLLDT for all residues within the HECT domain are appreciably high pLLDT > 70. This is consistent with our MD simulation, indicating the HECT domain effectively maintains its

structural integrity.

However, to understand the HECT domain flexibility, the AlphaFold PAE should be considered (see new **Figure S12a**, right side). This metric quantifies the level of confidence or uncertainties associated with each residue-residue distance in the protein structure, describing the spatial relationships between different domains within the protein construct. As depicted below, the AlphaFold predicts the relative position of the HECT domain with a high degree of uncertainty, indicated by a substantial error up to 12.0 Å. This suggests considerable flexibility of the HECT domain in relation to the E6AP core. Interestingly, this uncertainty in interdomain positioning aligns well with the fluctuations (up to RMSF = 10.0 Å) that the HECT domain exhibited over the course of MD simulation.

Although the E6AP HECT domain has been previously solved by crystallography (PMID: 10558980, (Huang, Kinnucan et al. 1999)) and the HECT domain is likely folded in our structure, the relative position between the body of E6AP and the HECT domain did not allow solution of the complex.

Potential weak HECT domain density

3. Two AF2-predicted structures were presented in this manuscript (**Fig S8a**: AF-Q05086-F1-model_v4 and **Fig S9a**: AF-Q05086-F1-model_v2.pdb). The first predicted structure was selected for MD simulation of apo-E6AP, and the second predicted structure was selected as a comparison with the experimental E6AP structure. What are the reasons for choosing different AF2 predicted structures? What will the results be if the two different AF2-predicted structures are both used for MD simulations and structural comparison? In addition, why the other AF2-predicted structures were not considered in the manuscript, such as AF-Q05086-F1-model_v1 and AF-Q05086-F1-model_v3?

Thank you for pointing this out and we can see how this was confusing. The origin of differences in file names is a result of the file download date. After comparing the file versions, we found that there was a 0.0 Å rmsd between the structures used in **Figure S12** and **Figure S13**. To maintain transparency in case there are unrealized differences between the files we kept the naming convention. To clarify this point, we amended the **Materials and Methods** section (line 678-680) to include: “For files AF-Q05086-F1-model_v1 and AF-Q05086-F1-model_v3, the version numbers were different due download date although the rmsd between the two structure files is zero. To maintain transparency, the file names were maintained.”

4. Page 7 and Fig S10b-d: To better characterize the stability of three sites of interactions between E6AP and E6, the authors are suggested to add a new panel in Fig S10 by providing the time evolution of the fraction/percentage of the E6AP-E6 native contact at each interaction site.

We thank the reviewer's suggestion. We included the new panel as **Figure 3b-d** and **Figure S14a-c** describing the time evolution of the fraction/percentage of the E6AP-

16E6 native contact at each interaction site.

5. Page 7 and Fig S10a: The authors calculated the time evolution of the RMSD values of each monomer in E6AP-E6-p53. This reviewer suggests the authors also calculate the RMSD values of the entire E6AP-E6-p53 complex, which can provide the overall structural stability of E6AP-E6-p53 complex.

We included the rmsd of the entire complex to provide a global view of the complex fluctuations. This new data is found in **Figure 3a**

6. There are several recent publications which are closely related to the content of this manuscript. Here are some examples, [Nat Rev Drug Discov 22, 127-144 (2023)], [Biophys J 121, 1704-1714 (2022)], [Nat Commun 12, 986 (2021)], [Pharmacol Res 177, 106128 (2022)], [Sig Transduct Target Ther 8, 92 (2023)]. These papers need to be discussed/cited in this manuscript.

We thank the reviewer for pointing out additional literature. We have included these citations in the manuscript and referenced them in the discussion.

7. Fig S8: There is a spelling error in "simulaiton", which should be corrected.

We thank the reviewer for catching the misspelling of "simulaiton"--we have corrected the spelling.

References:

Huang, L., E. Kinnucan, G. Wang, S. Beaudenon, P. M. Howley, J. M. Huibregtse and N. P. Pavletich (1999). "Structure of an E6AP-UbcH7 Complex: Insights into Ubiquitination by the E2-E3 Enzyme Cascade." Science **286**(5443): 1321-1326.

Martinez-Zapien, D., F. X. Ruiz, J. Poirson, A. Mitschler, J. Ramirez, A. Forster, A. Cousido-Siah, M. Masson, S. V. Pol, A. Podjarny, G. Travé and K. Zanier (2016). "Structure of the E6/E6AP/p53 complex required for HPV-mediated degradation of p53." Nature **529**(7587): 541-545.

Nakane, T., D. Kimanius, E. Lindahl and S. H. W. Scheres (2018). "Characterisation of molecular motions in cryo-EM single-particle data by multi-body refinement in RELION."

eLife **7**: e36861.

Punjani, A. and D. J. Fleet (2021). "3D variability analysis: Resolving continuous flexibility and discrete heterogeneity from single particle cryo-EM." Journal of Structural Biology **213**(2): 107702.

Ye, X., P. Zhang, J. Tao, J. C. K. Wang, A. Mafi, N. M. Grob, A. J. Quartararo, H. T. Baddock, I. Foe, A. Loas, D. L. Eaton, Q. Hao, A. H. Nile and B. L. Pentelute (2023). "Discovery of reactive peptide inhibitors of human papillomavirus oncoprotein E6." bioRxiv: 2023.2005.2025.542341.

Yip, K. M., N. Fischer, E. Paknia, A. Chari and H. Stark (2020). "Atomic-resolution protein structure determination by cryo-EM." Nature **587**(7832): 157-161.

Young, G., N. Hundt, D. Cole, A. Fineberg, J. Andrecka, A. Tyler, A. Olerinyova, A. Ansari, E. G. Marklund, M. P. Collier, S. A. Chandler, O. Tkachenko, J. Allen, M. Crispin, N. Billington, Y. Takagi, J. R. Sellers, C. Eichmann, P. Selenko, L. Frey, R. Riek, M. R. Galpin, W. B. Struwe, J. L. P. Benesch and P. Kukura (2018). "Quantitative mass imaging of single biological macromolecules." Science **360**(6387): 423-427.

REVIEWERS' COMMENTS

Reviewer #1 (Remarks to the Author):

I appreciated all the efforts done by the authors to provide further evidence on the strong affinity between E6 and E6AP. The revised manuscript deserves now to be published.

Note that the KD value of 22 micromolar determined by ITC (page 4, line 134) refers to the binding affinity between E6 (bound to LxxLL) and p53. Please correct this sentence.

Reviewer #2 (Remarks to the Author):

The determination of the 16E6, E6AP, P53 complex structure with supporting analysis of the interaction interfaces is a significant step forward in our understanding of the HPV complex that induces a number of epithelial cancers. The authors have addressed most of my concerns, however they still refer to the work as achieving atomic resolution on line 103 of page 3. This needs to be corrected to reflect the actual resolution of 3.3 Angstroms before publication. Although the HECT domain of E6AP was not pursued in the study either structurally or in other analyses, the authors have performed a significant amount of work on the remaining aspects of the complex to warrant publication. Additionally, proofs will hopefully catch the grammatical and redundant phrasings introduced into the manuscript during rewrite, such as those found on line 163, 169, and 244.

January 5th, 2024

Dear Reviewers,

Thank you once again for your insightful review of our manuscript, "**Structure of the HPV16 p53 Degradation Complex.**" Your valuable suggestions and commentary have significantly contributed to the enhancement of our manuscript from its original version. As a result, we are confident that the manuscript is now well-suited for publication in ***Nature Communications***.

Reviewer #1 (Remarks to the Author):

I appreciated all the efforts done by the authors to provide further evidence on the strong affinity between E6 and E6AP. The revised manuscript deserves now to be published.

Note that the KD value of 22 micromolar determined by ITC (page 4, line 134) refers to the binding affinity between E6 (bound to LxxLL) and p53. Please correct this sentence.

Thank you for catching this oversight. We have corrected the manuscript and it now reads "Compared to previously reported binding constants, we measured an approximately 10,000-fold higher affinity than the 2–4 μM K_d measured by SPR [34, 35] for the isolated E6AP LXXLL peptide."

Reviewer #2 (Remarks to the Author):

The determination of the 16E6, E6AP, P53 complex structure with supporting analysis of the interaction interfaces is a significant step forward in our understanding of the HPV complex that induces a number of epithelial cancers. The authors have addressed most of my concerns, however they still refer to the work as achieving atomic resolution on line 103 of page 3. This needs to be corrected to reflect the actual resolution of 3.3 Angstroms before publication. Although the HECT domain of E6AP was not pursued in the study either structurally or in other analyses, the authors have performed a significant amount of work on the remaining aspects of the complex to warrant publication. Additionally, proofs will hopefully catch the grammatical and redundant phrasings introduced into the manuscript during rewrite, such as those found on line 163, 169, and 244.

Thank you for catching our mistake in not removing the phrase "atomic resolution". It has been rectified in the manuscript and it now reads "To expand our understanding of this complex, we solved the ternary complex of 16E6 with the full length E6AP and the p53^{core} domain by cryo-EM to $\sim 3.3 \text{ \AA}$."

Thank you for taking the time to review our manuscript.

Sincerely,

Aaron H. Nile (aaronnile@calicolabs.com)

Principal Investigator, Oncology

Qi Hao, Ph.D. (qhao@calicolabs.com)

Principal Scientist 2, Associate Director of Protein Sciences